



# Carbon Budget Concept, its Deviations and the Equation: Climate Economics Perspective

Vito Avakumović [1]

[1]Center for Earth System Research and Sustainability (CEN), University of Hamburg, Grindelberg 5, 20144 Hamburg, Germany

**Correspondence:** Vito Avakumović (vito.avakumovic@uni-hamburg.de)

**Abstract.** The carbon budget concept is based on discoveries made at the end of the 2000s, claiming that global mean temperature (GMT) increase is roughly linearly dependent on cumulative emissions, with a proportionality metric named transient climate response to cumulative carbon emissions (TCRE). Since its emergence in natural science, the carbon budget concept has gained prominence as a tool for policymakers and climate economists alike. However, its usage in economic assessments

has been critiqued due to its inability to capture time-delay effects and TCRE changing with climatic conditions.

In this paper, we define the "perfect budget" equation as a purely linear relationship between GMT and cumulative emissions. Hereby, we distinguish two sources of deviations from the perfect carbon budget: emission scenario and climate state dependence. The former stems from the scenario choice under the fixed cumulative emissions, the latter from the change in TCRE with increased cumulative emissions. Introducing Green's function formalism in the context of the temperature response to

an emission pulse (pulse response), we provide reasoning behind time delay and possible scenario-dependent deviations and connect Green's formalism with the carbon budget equation. Using an optimization program, we show that extreme scenario-dependent deviations are relatively small compared to the overall GMT increase and even smaller when there are no more emissions after the year of optimization. We explain the behaviour of scenario-dependent deviations by the shape of the pulse response. Additionally, we show that the pulse response changes with climatic conditions, through which we further explain

carbon budget state dependency. Finally, we derive a generalized analytical carbon budget equation, which captures the state-dependent change of TCRE through a weak exponential GMT increase dependency on cumulative emissions. The analysis is done in FAIRv2.0.0, a simple climate model that includes climate feedback modifying the carbon cycle.

## 1 Introduction

The carbon budget concept, or the carbon budget approach, has gained prominence over the last decade for its ability to

determine allowable carbon dioxide emissions leading to a specific global mean temperature (GMT) increase. In essence, it assumes a direct link between the total cumulative carbon emissions and the temperature increase without the need to know the preceding emission pathway, making it both a powerful political and a convenient economic tool. Following the concurrent initial discoveries at the end of the 2000s (Allen et al. (2009), Matthews et al. (2009), Meinshausen et al. (2009), Zickfeld





et al. (2009)), the concept received wider recognition with its inauguration in the IPCC Fifth Assessment Report in 2013[1]

(AR5) (Stocker et al., 2013). Over the next 8 years, the carbon budget concept skyrocketed in its significance, such that in the IPCC Sixth Assessment Report (AR6), it arguably takes the main role as a policy recommended tool for limiting future climate change, as indicated by its gravity in the summary for policymakers (SPM), chapter D (Masson-Delmotte et al., 2021). To emphasize this claim, in AR6 Table SPM.2, "remaining carbon budgets" are identified explicitly – indicating how much carbon is left to emit to reach low-temperature targets, when assuming net-zero emissions afterwards. By and large, since its

emergence, the carbon budget has become "a staple of climate policy discourse": having paved the way for various discourses, from policy proposals and international climate justice discussions to financial recommendations and even climate activism arguments for the immediate abandonment of fossil fuels, to name a few (Lahn 2020).

In addition to its substantial policy implications, the carbon budget approach has an ever-increasing relevance in the field of climate economics. In the analytic climate economy (ACE) models that combine general production systems with climate

dynamics in an analytically tractable way, the carbon budget approach proves to be a convenient tool that simplifies the analytical approach (Dietz and Venmans, 2019). Since ACE models have a similar structure as their numerical counterparts, integrated assessment models (IAMs), there is a possibility that the carbon budget approach can make simple climate models used in integrated assessments redundant [2]. In decision-making theory, Held (2019) showed that it forms a bridge between two decision-making frameworks – cost-effectiveness analysis (CEA) and cost-risk analysis (CRA), the latter being robust under

the anticipated climate system learning event. The former, alongside the cost-benefit analysis (CBA) framework, has been the main tool in IPCC AR5, so the carbon budget approach, along with CRA, de facto saves numerous Working Group 3 results from dynamical inconsistency.

Formally, the carbon budget rests on the assumption that the GMT is increasing nearly linearly with cumulative emissions, regardless of the preceding carbon emission scenario. Hence, we define a "perfect budget" equation as follows:

$$T(t) = \lambda F(t), \tag{1}$$

where $F(t) = \int_0^t E(\tau)\mathrm{d}\tau$ stands for cumulative emissions, and $\lambda$ is the proportionality constant, named the transient climate response to cumulative carbon emissions (TCRE). The linear relationship emerges due to non-linearities counteracting each other: a concave temperature dependency on the atmospheric carbon content and a convex atmospheric carbon dependency on cumulative emissions (Matthews et al. (2009), Raupach (2013)). The former stems from the radiative efficiency saturation of

the atmospheric carbon, the latter from climate feedbacks weakening the natural carbon sinks.

However, Eq. (1) has been shown to be only an approximation, as the logarithmic effect takes over under higher climatic stress (higher $T$ and $F$). Leach et al. (2021) quantify the TCRE drop to approximately 10% per 1000 GtC. Additionally, Leduc et al. (2015) showed that constant TCRE is a good approximation for temperature response under low-intensity emission scenarios, while it overestimates the model's response to high-intensity scenarios; that reaffirms the necessity for TCRE to

decrease in order for the relationship in Eq. (1) to hold. In this paper, we define and label this change in TCRE with the

---

[1]It was not labelled explicitly as a budget but rather presented implicitly through the emphasis on temperature dependency on cumulative emissions (see Figure AR5 SPM.10).

[2]That is, if only carbon and no other non-$CO_2$ climate change drivers are examined.





changing climatic conditions (higher temperatures & cumulative emissions) as the climate *state-dependent carbon budget deviation*.

One could imagine a second source of deviation from the budget approach that stems only from the preceding emission scenario choice, and not the state of the system; we define and label it as an emission *scenario-dependent carbon budget*

*deviation*. The two introduced carbon budget deviations are inherently independent of each other. The aforementioned state dependence affects the proportionality constant (TCRE) depending on the cumulative emissions $F$. The emission scenario-dependent carbon budget deviation, however, implies the possibility of achieving a different temperature $T$ following the same amount of cumulative emissions $F$. The acquired difference can then only depend on the preceding emission scenario choice. Previous literature, utilizing the high-complexity climate models (ESMs), argues in favour of scenario independency (Gillett

et al., 2013). However, the problem with testing the emission scenario effects with ESMs is that they are computationally costly, which means only a limited set of stylized emission pathways are examined. This problem was tackled by Millar et al. (2016) by forcing the simplified, globally aggregated climate model under numerous emission scenarios. However, we assert that the extreme cases of maximally possible scenario-dependent carbon budget deviations are yet to be investigated and scrutinized.

Another often overlooked issue with the carbon budget approach is the absence of its consolidated definition regarding

the timescale it operates on. If the carbon budget approach is unambiguously interpreted as in Eq. (1), it would suggest an immediate temperature increase in response to cumulative emissions. In reality, there is always some time lag between the input and the reaction of the climate system. The inability of the carbon budget approach to account for temperature delay was critiqued by Traeger (2021) in the context of ACE. Furthermore, Peters (2018) argues that multiple carbon budgets can be deducted from the same emission scenario, depending on how the budget is defined.

This paper aims to define and scrutinize both the scenario- and state-dependent deviations of the carbon budget approach.

We identify the maximally possible realistic carbon budget scenario-dependent deviations by using the optimization program that maximizes and minimizes temperature in a specific year for fixed cumulative emissions. Through the optimization scheme, the extreme cases are tested; hence, we obtain the upper bound for the scenario-dependent deviations while encompassing the possible time-delay effects simultaneously. We argue that the two effects are interlinked, given that we show that scenario-

dependent deviations stem from the time-delayed response of the system.

We propose using a temperature response to an emission pulse (e.g. pulse response) in the context of Green's function formalism, offering a reinterpretation of the "perfect budget" equation (Eq. (1)), with the idea that it can help to provide the intuition behind the carbon budget deviations. We show that the "perfect budget" equation is only a special case of Green's function equation. Furthermore, Using the Green's approach in the above-mentioned optimization program and comparing it

to the full-fledged model results, we confirm its ability to capture scenario-dependent effects; moreover, its inability to capture state-dependent effects is also revealed.

With this knowledge, we dive into the causes of the deviations through the lens of the pulse response shape. We show how the shape of the pulse response determines possible scenario dependency. Further, we show why the utilization of a non-changing pulse response leads to the inability to capture state dependency – since the pulse response shape and magnitude also change

over time. Finally, we utilize the fact that pulse response changes with climate conditions to derive the state-dependent carbon





budget equation, and show that it can emulate the full-fledged model up to five times higher precision than a regular linear carbon budget relationship.

Overall, this paper aims at an audience from climate economics because it deals with issues that we find vital for that branch: carbon budget, its deviations, and possible utilization of the carbon budget approach in integrated assessments or ACEs. From a natural science perspective, it can be categorized into the body of literature that deals with the conceptual mechanisms behind the carbon budget approach (Raupach (2013), MacDougall and Friedlingstein (2015), Allen et al. (2022)). The difference from the listed literature is that we do not look for the reasoning within the climate model itself, but instead draw conclusions from the output of the model.

Following the introduction, the article is arranged as follows. In Sect. 2, we introduce the models and connect Green's framework with the carbon budget equation. In Sect. 3, we deal with scenario dependency. We introduce the optimization procedure, followed by the resulting scenario-dependent deviations. In Sect. 4, we detect the source of the generated deviations through the lens of the pulse response (Green's) function. In Sect. 5, we use the fact that pulse response changes with climatic conditions to derive the state-dependent carbon budget equation. In Sect. 6, we reflect on the findings and conclude the work.

## 2 Models

This paper utilizes a climate model that includes climate feedback on the carbon cycle. Because we use an optimization procedure to generate carbon budget deviations, the model has to be chosen to be computationally cheap. We will distinguish and use two models for inspecting the carbon budget deviation, the full-fledged model and the Green's function model. While the former is sufficient for inspecting the carbon budget scenario dependence, the latter mathematically formalizes the carbon budget approach and deepens the understanding of the deviations. The native FAIR language is Python; while in this paper it is incorporated in GAMS.

### 2.1 Full-Fledged Fair

By the full-fledged model, we mean the FAIRv2.0.0 model as prescribed by Leach et al. (2021). The model belongs to a class of simple climate models (SCMs), used in integrated assessment models due to their computational affordability. Dietz et al. (2021) suggested it is currently the top choice SCM for climate policy analyses[3] We follow their advice with the two features of FAIR being crucial for this paper. The first is the above-mentioned ability to correctly capture the temperature response following one carbon emission pulse, i.e. pulse response (Millar et al., 2017). The second feature is its ability to incorporate climate feedback, with one of the effects being a correct representation of change in pulse response with changing climatic conditions.

In essence, the FAIR model is a SCM designed to emulate the gas dynamics of different radiative forcers and their effect on the global mean temperature. Because we are interested only in the deviations from the carbon budget, we leave out the

---

[3]They actually argue for the FAIR version 1 given in Millar et al. (2017). However, Leach et al. (2021) argue that the second version is built to emulate the first, with the advantage of being more user-friendly and improved temperature dynamics.





non-$CO_2$ forcers of the analysis, utilizing only the carbon cycle system and its radiative forcing dynamics. The brief model overview follows, while the full model description and equations are provided in the Appendix.

FAIR's carbon cycle consists of four carbon and three temperature boxes (compartments), whose total sum gives the atmospheric carbon concentration inventory and global mean temperature anomalies, respectively. The preindustrial state is equal to all compartments' values set to zero. Each carbon compartment has an associated decay timescale which dictates the dissipation of the carbon content into the shared permanent pool that represents the global natural carbon sink. Along with the global temperature increase, the sink's increased content increases creates a feedback mechanism. Intuitively, the feedback mechanism mimics the effect of weakening the carbon sinks (absorbing carbon saturates the sinks), while the temperature increase weakens the sinks by, for example, decreasing the carbon solubility in the ocean. With weaker sinks, decay timescales increase, and so does atmospheric carbon retention time.The atmospheric concentration gives rise to the radiative forcing by combining a logarithmic and square root term. The resulting forcing has the dynamics in temperature boxes equivalent to carbon emissions' dynamics in the carbon boxes.

We implement the standard (default) parametrization provided by Leach et al. (2021) for the temperature and the carbon cycle module. The standard parametrization can be found in the Appendix.

## 2.2 Green's Function Framework

### 2.2.1 Green's Function Formalism

Green's model is, in fact, one equation motivated by Green's function formalism. In essence, a Green's function $f_g(t - \tau)$ is a specific function unique to a set of linear differential equations $Lx(t) = y(t)$, where $y(t)$ is the input forcing and $x(t)$ is the state variable that changes according to the forcing and the linear operator $L$. The advantage of Green's function is that it acts as a "propagator" from the input variable (external forcing) to the output variable (change in state variable) in such a way that it is possible to circumvent the differential equations with just one equation that reads as $x(t) = \int_{t_0}^{t} y(\tau) f_g(t - \tau) \mathrm{d}\tau$.

Using the same formalism, we propose Green's equation in the context of global mean temperature dynamics. We are interested in inspecting the temperature dependence on the emission pathway, where the latter and former take the abovementioned role of input and output variables, respectively. Hence, we propose the following equation imitating Green's equation formalism:

$$T(t) = \int_{t_0}^{t} E(\tau) f_g(t - \tau) \mathrm{d}\tau. \tag{2}$$

The output variable is the global mean temperature change $T(t)$, and the input (forcing) variable is the emissions $E(t)$. Green's function $)f_g(t - \tau)$ modifies the contribution to a current temperature $T(t)$, stemming from the emission in the past $E(\tau)$. According to Eq. (2), the temperature in time $t$ will depend on each emission contributing at time $\tau$ prior to $t$, with the effect modified with Green's function $f_g$ dependent on how far the emission year $\tau$ is from $t$, hence $f_g(t - \tau)$. Essentially, it is an integration scheme that counts temporarily modified temperature contributions of each emission pulse, going backwards from moment $t$. Hence, it captures the time delay effects of translating emissions to temperature change.





### 2.2.2 Pulse Response as Green's Function

To use Eq. (2), must opt for a shape of a Green's function $f_g$. Following the proposed definition, we choose it to be a temperature
evolution response following the 1 GtC emission pulse, or simply, the "pulse response". Therefore, in this paper, the terms
"Green's function", "pulse response" or "temperature evolution following the emission pulse" all have the same meaning.
Pulse response experiments are one of the generic experiments when inspecting climate models. We follow the traditional way
(Joos et al. (2013), Millar et al. (2017)) of generating the pulse response by adding a pulse on top of a constant atmospheric
concentration background[4].

In previous literature, authors have tested the pulse response following the 100 GtC emission injection. Compared to its
GtC counterpart, the qualitative shape of the pulse response stays the same for a 1 GtC injection, with the magnitude
scaled down accordingly. Generating the pulse response (Green's function) is done by utilizing the full-fledged FAIR model,
as follows.

Until 2020, the full-fledged model was forced by RCP6.0 carbon emission scenario provided by the RCMIP protocol
(Nicholls, 2021), starting from the preindustrial era. In 2020, the model concentration level of 402 ppm is reached and, from
that year, kept constant, while the emissions necessary to keep that concentration level unchanged are diagnosed. One ad-
vantage of the GAMS programming language is that it allows this procedure without rearranging equations, as is necessary
for Python (the original language of FAIR model). Instead, we change the role of emissions and concentrations from input
to output variables and vice versa. After the emissions required to keep the concentration constant are generated, we run two
experiments from the year 2020: one with the generated emissions only and one with 1 GtC extra added in 2020. Thus, pulse
response (Green's function) is generated by subtracting the temperature evolution of the two runs. The Green's function $f_g$
generated in such a manner, utilized in Green's model (Eq. (2)), is depicted in Fig. 4 in blue (labelled as Green1.

### 2.2.3 Carbon Budget Equation in the Context of Green's Formalism

Next, we inspect the connection between Green's function (Eq. (2)) and the carbon budget suggested by Eq. 1. As a first test of
Green's approach, we show that the "perfect budget" equation is just a special case of the former. Furthermore, the viability of
Green's approach is numerically tested in the next section, compared to full-fledged FAIR in the context of scenario-dependent
deviations.

Essentially, the "perfect budget" equation suggests an immediate temperature response to (cumulative) emissions, and that
that response does not change in time or with climatic conditions. That implies that the pulse response introduced in the
previous subsection should also be a constant function. In Fig. 4, it is plotted in a dashed black line: the temperature response
to an emission pulse has an immediate jump following the emissions, and it does not change in time, as the "perfect budget"
implies.

Formally, a "perfect budget" temperature response can be interpreted as a Heaviside function $\Theta(t)$ multiplied by a constant.
If the carbon budget framework is consistent, it has to be equal to TCRE. Hence, we set it equal to the central estimate of

---

[4]One possible way would be adding an emission pulse on top of a prescribed emission pathway.





TCRE from Leach et al. (2021), $\lambda = 1.53 \cdot 10^{-3}$ °C GtC$^{-1}$. With that, we propose the "perfect budget" temperature response $f_g^0$ to have the following mathematical form:

$$f_g^0(t-\tau) = \lambda\Theta(t-\tau) = \begin{cases} 0 & t < \tau \\ \lambda & t \geq \tau \end{cases}, \tag{3}$$

where $\tau$ is the timing of the emission pulse and is equal to 0th year in Fig. 4. Clearly, there is no temperature response before the pulse; therefore, the Heaviside function has a momentary jump in the time of the emission pulse.

Now we show that Green's formalism can be considered an analogue to the carbon budget approach. We assume the "perfect budget" temperature response (Eq. (1)) as a Green's function and insert it into the Green's equation (Eq. (2)), and then the claim is easily proved:

$$T(t) = \int_{t_0}^{t} E(\tau)f_g^0(t-\tau)\mathrm{d}\tau = \int_{t_0}^{t} E(\tau)\lambda\Theta(t-\tau)\mathrm{d}\tau = \lambda\int_{t_0}^{t} E(t')\mathrm{d}t' = \lambda F(t).$$

The derived equation is precisely the carbon budget equation, as given in the introduction. We can conclude that if the temperature response always had the same (constant) shape as the dashed line in Fig. 4, the carbon budget would have deviations and no time delay – each unit of carbon emission momentarily adds to the warming equally, regardless of when it was emitted.

However, as shown in Fig. 4, the FAIR-generated pulse response (blue line) is not a constant function. Hence, Eq. (1) does not perfectly hold, and Eq. (2) can be interpreted as a generalized carbon budget equation. Because of the (non-constant) shape of the generated pulse response, Eq. (2) generalizes the carbon budget equation in a way that enables time-delay effects that lead to possible emission scenario dependencies. The following section will test Green's framework against the full-fledged model. Furthermore, Sect. 4 thoroughly scrutinizes the connection between the pulse response shape and possible carbon budget scenario-dependent deviations.

Before going further, we briefly go back to the formalistic introduction of Green's framework from the aspect of carbon budget state-dependent deviations. By proposing Eq. (2) and using FAIR-generated Green's function, we assume that FAIR is a set of linear differential equations. This is false because non-linearities arise in the carbon cycle through the climate feedback parameter and in the logarithmic dependence of radiative forcing on atmospheric concentration. This effectively means that while Green's function can capture time-delay effects, it cannot capture the effects of climate state change on the carbon budget approach. The effect is visible when comparing full-fledged and Green's model optimization runs with higher cumulative emissions. Nevertheless, Sect. 4 shows how state dependency can be detected through the changing pulse response. Hence, pulse representation helps us understand changing TCRE with climatic conditions and, finally, derive the new, reinterpreted carbon budget equation that captures state-dependent deviations.


## 3 Scenario-Dependent Deviation

### 3.1 Method

#### 3.1.1 Minimization/Maximization Scheme

To test the possible scenario-dependent carbon budget deviations, the optimization program is formulated as follows:

$$(\text{Max}, \text{Min})[T(t^*)] \quad \text{s.t.} \int_{t_0}^{t^*} E(t)\mathrm{d}t = F_{\text{tot}}, \left|\frac{\mathrm{d}E(t)}{\mathrm{d}t}\right| \leq k, E(t) \geq 0, E(t_0) = E_0. \tag{4}$$

Using the full-fledged and Green's model independently, the program maximizes (minimizes) the temperature variable in a given specific optimization year $t^*$. The generated minimal $T_{\min}(t^*)$ and maximal $T_{\max}(t^*)$ temperatures provide the upper and lower bound for possible temperatures under the given constraints, elaborated in the following paragraphs. The maximal 220 possible scenario-dependent carbon budget deviation $T_{\text{d}}$ is calculated by subtracting the two boundary temperatures, $T_{\text{d}}(t^*) = T_{\max}(t^*) - T_{\min}(t^*)$.

In the optimization program (Eq. (4)), the emission pathway takes the role of the free control variable, except in the fixed initial condition $E(t_0) = E_0$. Hence, the novelty of testing the scenario independence with the optimization program is that the emission pathway is generated, instead of being assumed as an input by the user. To both avoid trivial solutions and keep the 225 generated emission pathways within what we deem as realistic, three boundary conditions are implemented.

The first boundary condition sets the total cumulative emissions at the year of optimization $t^*$ to a fixed value $F_{\text{tot}}$, counting from the initial year $t_0$. The condition restricts the emissions from diverging by keeping them within realistic boundaries. More importantly, it ensures that the deviation from the carbon budget stems only from the difference between the emission pathways (fixes the cumulative emissions to be equal at the end of both the minimization and maximization run). In our experiment, the 230 chosen $F_{\text{tot}}$ values are 416, 600, 800, and 1000 GtC. The first one (416 GtC), in addition to historical cumulative emissions, amounts to 1000 GtC since the preindustrial era, which approximately corresponds to the carbon budget of keeping the global mean temperature below 2°C with 67% probability, as suggested by the IPCC report (Masson-Delmotte et al. (2021), Table SPM.2). In our run, however, 2 °C will not be reached with those cumulative emissions since the effects of other non-$CO_2$ radiative forcers are left out. Other values for $F_{\text{tot}}$ are chosen generically to test the effect of higher cumulative emissions on 235 scenario-dependent deviations.

The second boundary condition gives the upper bound on the rate of change of emissions per year, effectively setting the allowed absolute slope of the emission pathway to be less than or equal to a prescribed value $k$. Hence, a trivial solution (e.g. emitting all of the emissions in one year) is avoided. Furthermore, $k$ is chosen in a way to be politically relevant. It is restricted to have the upper bound of 1 GtC/yr$^2$, roughly corresponding to the emission reduction rate if the annual emissions were 240 linearly reduced to zero between 2020 and 2030.

The combination of the restriction on $k$ with the $F_{\text{tot}}$ restriction will affect the run's feasibility. The higher the cumulative emissions and the lower the $k$ is, the less likely the run is feasible. We can easily visualize that if we keep in mind that cumulative emissions are the surface area below the emissions path. First, remember that we start the optimization in the year





$t_0$ with the corresponding fixed initial emissions $E_0$. Now, let us assume that we want to reach some $F_{\text{tot}}$ at the year $t^*$; we
can immediately conclude that $k = 0$ is feasible for only one combination of $E_0$, $t^*$ and $F_{\text{tot}}$, that is: $F_{\text{tot}} = E_0(t^* - t_0)$. If we
additionally require that we reach net-zero by $t^*$, the run becomes infeasible since there ought to be a slope to reach that goal.
If we allow the emissions to increase (decrease) fast enough with a larger $k$ value, the run becomes feasible again (for fixed
$F_{\text{tot}}$).

     Additionally, the feasibility limiting value of $k$ will correspond to the run where both $T_{\max}(t^*)$ and $T_{\min}(t^*)$ are equal; hence,
the scenario-dependent carbon deviation $T_{\text{d}}(t^*)$ is zero for that specific $k$. This is to be expected since for the limiting value
of $k$, only one emission pathway is feasible under the specified $F_{\text{tot}}$. Therefore, only one emission pathway can be generated,
equal for the minimizer and the maximizer. Following the same logic, if we are slightly above the limiting $k$, the deviations
will always be small, regardless of other variables (e.g. $F_{\text{tot}}$). With the higher $k$, the range of possible pathway combinations
increases, and so does $T_{\text{d}}$.

The last boundary condition does not allow for negative emissions. While there are certain indications that they ought to be
employed to meet low-temperature targets, it is unclear how well FAIR fares with negative emissions and if it makes sense to
use the model for that purpose.

### 3.1.2    Two Cases of Carbon Budget Interpretation

To add another layer to the discussion on how the carbon budget can be interpreted, we distinguish two additional cases that
differ depending on how we emit after the time of interest (in our case $t^*$).

     We address the first carbon budget interpretation as a "net-zero budget" case, which corresponds to the situation in which all
of the carbon has been emitted up until the time point of interest, and there are no other emissions afterwards. This interpretation
coincides more with a carbon budget in a literal sense of budget, which states how much carbon is left in-store for reaching
specific targets. Henceforth, we attribute it to the policy-relevant domain.

On the other hand, we can distinguish the so-called "open budget" case. In this case, the net-zero requirement at the year of
interest does not need to be fulfilled, as we are only interested in the momentary relationship between the current cumulative
emissions and current temperature increase. This interpretation can be attributed to the carbon budget approach, as seen through
the lens of climate economics through the instantaneous relationship between temperature and cumulative emissions, as given
in Eq. 1. As it will be shown, the scenario-dependent carbon budget deviation strongly differs in magnitude, depending on
which of the two cases we are looking at.

     To see how the two above-mentioned interpretations of the carbon budget differ in regard to the scenario-dependent devia-
tions they generate, we introduce two additional scenarios. The scenarios an follow identical mathematical structure as given
in Eq. 4, except for a different boundary condition on emissions at $t^*$.

     The first scenario (S1) is associated with the "net-zero" interpretation. In addition to boundary conditions given in Eq. 4, in
S1 we require emissions to reach zero by the year $t^*$ and stay zero from there onwards ($E(t \geq t^*) = 0$). Conversely, the second
scenario (S2) associated with the "open budget" has no extra requirements on top of the ones given by Eq. 4.





The additional constraint on the emission pathway negatively affects the feasibility mentioned above. Therefore, compared to S1 ("net-zero") , S2 ("open budget") has more possible emission pathway combinations available, which means that a higher $T_d$ is to be expected.

### 3.1.3 Deviation Time Evolution

Finally, the optimization procedure (Eq. (4)) calculates the extreme case of scenario-dependent deviations in one specific year $t^*$ only. To see whether these deviations are persistent in time, we design one extra experiment unique to S1. For each of $k$ specified in the setup above, we let the system evolve for the next 50 years following the optimization year ($t^* = 2070$), without adding new emissions and with it, allow $T_d(k)$ to evolve freely in time, whilst keeping cumulative emissions on the same level. In that way, we can see how the scenario-dependent deviation obtained in $t^*$ changes in time.

### 3.1.4 Run Configuration

Preceding the initialization in the optimization program, the FAIR model was historically forced from the preindustrial period (the year 1850) until year 2020 under the same setup as for the Green's function generated, described in 2.2.2. The historical run is dynamically separated from the optimization run since, in the former, emissions are prescribed, not generated by the program. The meeting point is the year 2020, where the historical run's variables' values are translated into initial conditions of the variables of the full-fledged optimization run. Hence, $t_0 = 2020$ in Eq. 4 and the initial emissions value of the optimizer run ($E_0$) are fixed to match the historical emissions in 2020. The initial temperature at $t_0$ is 0.96 °C, with the associated historical cumulative emissions counting 584 GtC.

Green's model's run requires an additional modification to make it comparable with the full-fledged model. As we can see in Eq. (2), Green's approach responds only to emissions that we feed it within the integral. That means that in the optimization run, which starts at $t_0$, it cannot capture the temperature response stemming from the previous, historical emissions. Conversely, this is not a problem for the full-fledged model since that "leftover" response is fed into the initial conditions of the run. To overcome this in Green's approach, we add the "temperature leftover" parameter $T_{\text{left}}(t)$ to Eq. (2), so it takes the form of $T(t^*) = \int_{t_0}^{t^*} E(\tau) f_g(t^* - \tau) \mathrm{d}\tau + T_{\text{left}}(t^*)$. The temperature leftover term is easily generated using the full-fledged model, with the emissions following the historical run and then set to zero from 2020 onwards, where $T_{\text{left}}(t)$ is diagnosed as the temperature evolution in the years when we stopped emitting.

### 3.2 Results

#### 3.2.1 "Net-zero Budget" (S1) Deviation

In the left panel of Fig. 1, the generated maximal (red) and minimal (light blue) temperatures for "net-zero budget" case are shown. The lower bound of $k = 0.4 \, \text{GtC/yr}^2$ is close to the feasibility limit, detectable by the minimal and maximal temperature proximity. Furthermore, we can notice that both Green's (dashed line) and full-fledged (solid line) models follow a similar trend, with the former being shifted upwards due to the state-dependence of the carbon budget, discussed later. Finally, we can see



that the maximal and minimal temperature (if focusing solely on a full-fledged approach) varies around 1.58 and 1.59 °C, as opposed to the anticipated 2 °C following the IPCC suggested cumulative emissions amount. The difference is due excluding

non-$CO_2$ forcers from our analysis and the parametrization choice.

The maximal scenario-dependent "net-zero budget" deviation $T_d$ depending on the slope restriction $k$ is plotted in the right panel in light pink as S1-FF for full-fledged and S1-G for Green's model, respectively. The deviation increases with the $k$. Furthermore, the deviation's magnitude is relatively small compared to the associated temperature increase. For the highest slope allowed ($k = 1\,\mathrm{GtC/yr^2}$), this setup's most significant possible deviation is approximately 1.5% of the overall temperature

increase. The upper bound of $T_d(k)$ is about 0.025 °C, accounting for only around one-quarter of a tenth of a temperature degree.

Finally, the deviation derived from Green's approach is in the same order of magnitude as the deviation derived from the full-fledged approach, and the trend behavior between the two is comparable. This justifies using Green's function to explain the source of scenario-dependent deviations in Sect. 4. The slight shift between the full-fledged and Green's output is due to

the fact that we use a constant Green's function, and that effect will be even more pronounced with higher $F_{tot}$ allowed (Fig. 3); to be discussed in Sect 4.

Figure 2 tests the time persistency of detected $T_d(k)$. Figure represents the time evolution of $T_d(k)$ following the optimization year $t^*$, with different shades of red depicting the $k$ range as given in the abscissae in Fig. 1. The initial values in the year 2070 correspond to the values of $T_d(k)$ of the "net-zero" case (Fig. 1, right panel, S1-FF). The scenario-dependent deviation,

generated for $t^* = 2070$, dies out in time if no additional carbon is added due to the system coming to a dynamic equilibrium. This shows that the maximal deviations generated by the optimization program are only temporary and, in combination with their relatively small magnitude, one could safely ignore them.

### 3.2.2 "Open Budget" (S2) Deviation

In Fig. 1 are shown, the optimization run results for the "open budget", with the equivalent setup as for the "net-zero" (Subsect.

3.2.1).

Focusing on the left panel, we see that the generated maximal temperatures (red) are the same for both cases. Furthermore, we can see that relaxing the condition on $E(t^*)$ leads to a significantly lower minimum temperature for the "open budget" (dark blue). The lower minimal temperature results from emissions allowed to stack up at the end of the run (not being pushed towards zero as the "net-zero" condition requires).

Resultingly, there is a larger gap between the maximal and minimal generated temperature. Therefore, as shown on the right panel, the "open budget" case shows a significantly larger scenario-dependent carbon budget deviation (dark pink) than the net-zero counterpart (light pink). For the highest allowed $k$. the deviation is around 0.095 °C.

Finally, we can see that the feasibility limiting $k$ is lower in the "open budget" than in the "net-zero" case – also a consequence of a non-constrained $E(t^*)$. Hence, because the signal is much clearer, we opt for a "open budget" approach to show

how $T_d(k)$ changes with the increasing total cumulative emissions $F_{tot}$. In Fig. 3, the results of four different optimization runs

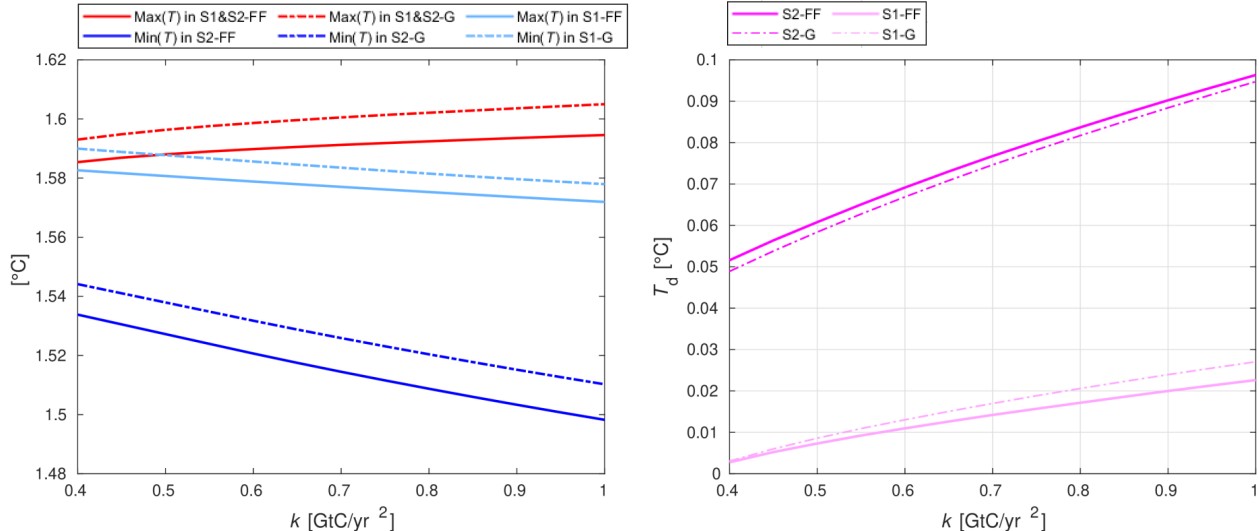

**Figure 1.** The left panel shows the maximal and minimal temperatures depending on the maximal emission slope allowed ($k$), generated by the optimization program (Eq. (4)) with $F_{\text{tot}} = 416$ and $t^* = 2070$, while the right panel gives the associated scenario-dependent deviations given by subtracting Min($T^*$) and Max($T^*$). Different lines represent different run setups with a combination of abbreviations defining a specific run. The abbreviations used are S1 and S2 for the "net-zero budget" and "open budget" interpretations of the budget, and FF and G for full-fledged and Green's model, respectively. For example, *Max(T*) in S1-G* represents the minimal generated temperature by Green's model for the "net-zero" case run.

in the "open budget" approach are presented. The optimization year is $t^* = 2090$[5], and cumulative emissions take the whole range introduced in the methods section. Three main effects can be distinguished.

First, $T_{\text{d}}(k)$ increases with higher cumulative emissions. A comparison of the first and the last right panels shows that the deviation increased by roughly 60%, associated with the $F_{\text{tot}}$ increase from 416 GtC to 1000 GtC. In the most extreme case 345 with associated $F_{\text{tot}} = 1000$, a deviation of $\sim 0.15$ °C is detected.

Second, the choice of the optimization year $t^*$ seems not to affect the deviation, if infeasibility effects are ignored. Namely, one can argue that a difference between two examples in the lowest $k$ choices can be detected. This is where the infeasibility effect occurs: the $t^* = 2090$ case has a slightly higher limiting $k$ close to 0.1 GtC/yr$^2$. In comparison, the limiting $k$ for $t^* = 2070$ is lower – visually depicted with the meeting point of the corresponding blue and red lines on the left. The two are 350 nearly identical from from roughly $k = 0.15$ GtC/yr$^2$ onwards. Hence, one can conclude that the deviation is not a function of the optimization year.

Third, the gap between Green's model and its full-fledged counterpart's maximal and minimal temperatures steadily increases with higher cumulative emissions $F_{\text{tot}}$ (left panels). This shift unambiguously indicates the inability of Green's model to capture non-linearities, as discussed in the method section. Additionally, the $T_{\text{d}}(k)$ difference between the two models also

---

[5]The optimization year is pushed farther to increase the feasibility and signal clarity





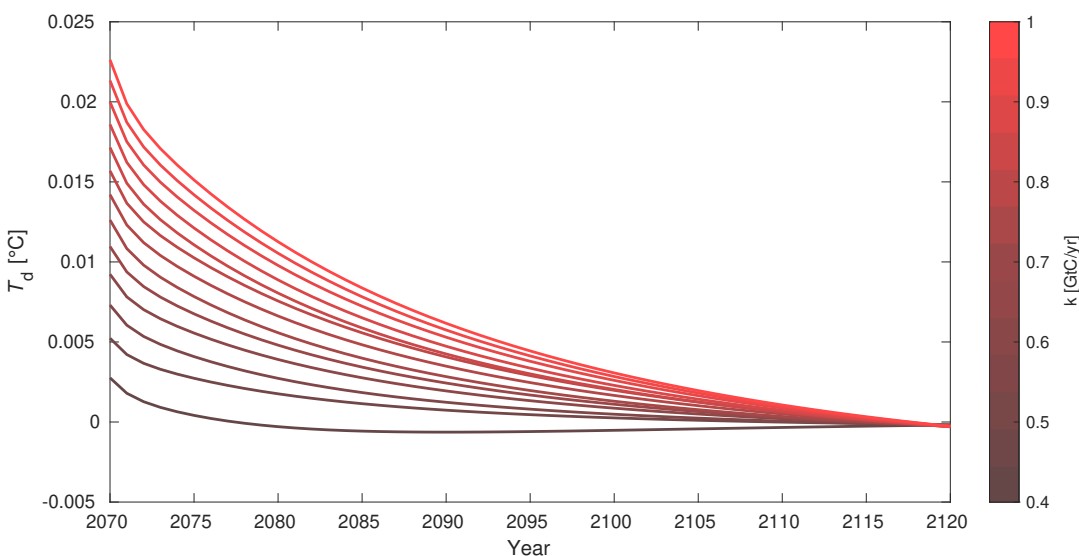

**Figure 2.** Temporal evolution of the "net-zero" case scenario dependent carbon budget deviation, 50 years following the optimization year. The colors represent carbon budget deviation corresponding to different highest emission slopes allowed in the optimization process $k$, with the darkest red being the lowest allowed slope (0.4 GtC yr$^{-1}$) and the brightest red the highest allowed slope (1 GtC yr$^{-1}$).

increases with higher $F_{\text{tot}}$, albeit to a lesser extent. This effect can be attributed to the widening gap between the maximal and minimal temperature of the full-fledged approach, which increases its $T_{\text{d}}(k)$ to a larger extent when compared to Green's model. As we show in next section, the last two findings can be interpreted through the lens of the pulse response function, which changes in magnitude and shape under different climatic conditions.

## 4 Pulse Response as a Deviation Source

In the previous section, we have demonstrated extreme cases of scenario dependency.irstly, we have shown that we can emulate the full-fledged model's generated scenario-dependent deviations using a pulse response as Green's function (Eq. (2)). Hence, explaining the sources of scenario-dependent deviations by inspecting the pulse response's shape is justified. Secondly, we have shown that the gap between full-fledged and Green's generated temperatures increases with higher cumulative emissions. As we will now show, this is because we use a constant pulse response in Green's approach, while in reality, the pulse also

changes with higher climatic conditions.

In this section, we contextualize the carbon budget approach and its deviations through the lens of the temperature response to an emission pulse. This behaviour of the pulse response has broader implications on other (simple) climate models and to which extent they adhere to the carbon budget approach.



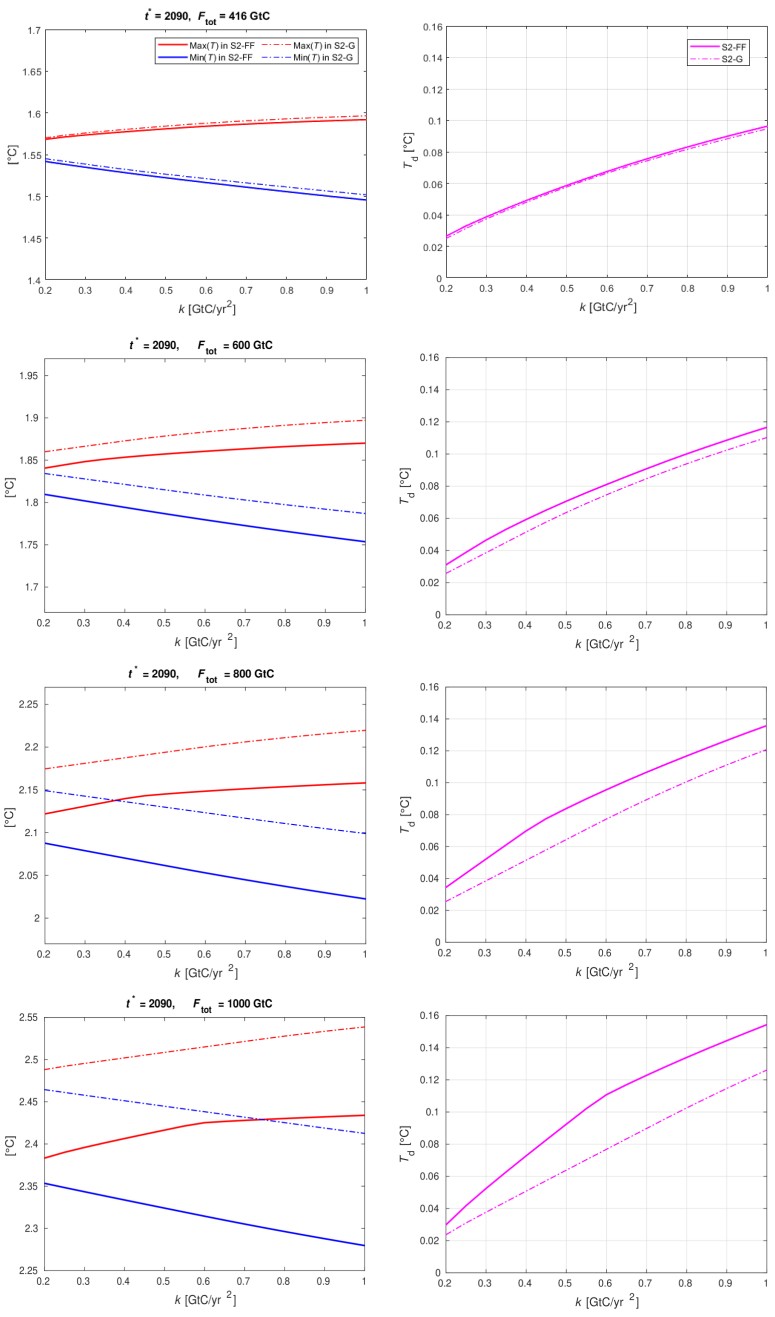

**Figure 3.** Left panels: Maximal (red) and minimal (blue) temperature dependent on k under the "open budget" case (S2), set up such that $t^* = 2090$ and $F_{\text{tot}} = (416, 600, 800, 1000)$ GtC. To get total cumulative emissions from preindustrial era, one needs to add 584 GtC to $F_{\text{tot}}$ values, accounting for the emissions prior to $t_0$. The panels are ordered by the magnitude of the associated $F_{\text{tot}}$. Note that the y-axis domains are chosen to have the same relative interval of 0.3 °C but different absolute values. This way we shift the focus on the deviations and differences between Green's (G) and full-fledged (FF) model instead of the acquired absolute temperatures. Right panels: corresponding scenario dependent carbon budget deviation $T_{\text{d}}(k)$. Solid lines represent full-fledged, and dashed lines Green's approach.



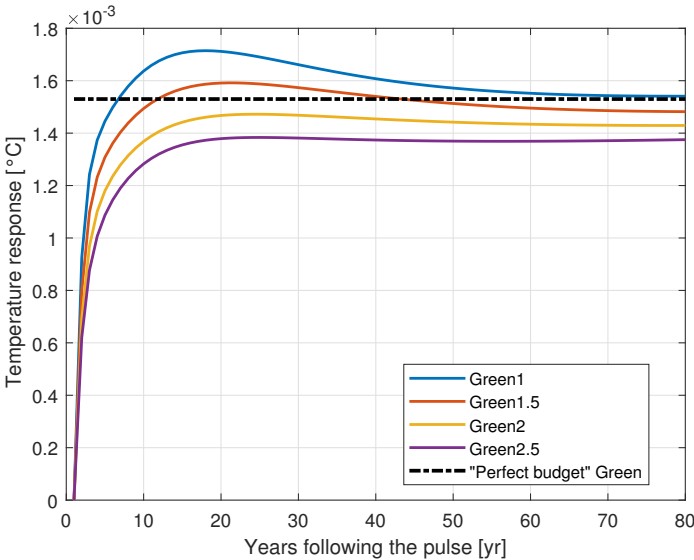

**Figure 4.** Temperature evolutions in response to 1 GtC emission pulse for different climatic conditions (colored) and the temperature response implied by Eq. (1) (black dashed). The numbers in the names indicate the global mean temperature anomaly under which they were generated. Green1 is a pulse response used as a Green's function (Eq. (2)) in the optimization runs (Sect. 3). Green1.5 corresponds to the pulse generated in the year 2055 of the RCP6.0 scenario run, with the corresponding cumulative emissions $F_{2055} = 959\,\mathrm{GtC}$, global mean temperature anomaly $T_{2055} = 1.5\,^\circ\mathrm{C}$, and a background atmospheric concentration $C_{2055} = 485\,\mathrm{ppm}$. Accordingly, Green2 is generated with the starting year 2078, $F_{2078} = 1317\,\mathrm{GtC}$, $T_{2078} = 2\,^\circ\mathrm{C}$, $C_{2078} = 577\,\mathrm{ppm}$, and Green2.5 with $F_{2100} = 1657\,\mathrm{GtC}$, $T_{2100} = 2.5\,^\circ\mathrm{C}$ and $C_{2100} = 660\,\mathrm{ppm}$.

## 4.1 Pulse Response Shape as a Scenario (In)Dependency Indicator

To pinpoint the source of the deviations, we first briefly revisit the discussion from Subsect. 2.2.3. As shown, the perfect carbon budget (Eq. (1)) implies that the pulse response is a constant step-function. However, the pulse response function used in Green's model, depicted in Fig. 4 in blue, labelled as "Green1", shows a dynamical temperature response.

In contrast to a constant step function, the initial response at the year of the emission pulse is zero. Then it starkly increases until reaching a maximum value of approximately $1.7 \cdot 10^{-3}\,^\circ\mathrm{C}\,\mathrm{GtC}^{-1}$, 17 years following the pulse[6]. Furthermore, following the peak, there is a slow relaxation of the response, which slowly reaches a constant response later in time. Together, the non-instantaneous response followed by the sudden temperature increase and the temperature response peak provides a hint for understanding the detected carbon budget scenario deviations in Figures 1 and 3. In contrast, the relaxation following the temperature peak explains the diminishing of the deviation later in time, as shown in Figure 2.

---

[6]Note that using the FAIRv1.0.0 in Millar et al. (2017), the pulse peak is reached at year 12. The difference is because in the newer FAIR version the third temperature box was introduced, whilst previously there were only two. The slight shift of the maximum does not affect the brought up conclusions.





To get an intuition of the deviations and how they are connected to the pulse, we consider an extreme example. Let us imagine that all of the emissions are injected in one year. Total cumulative emissions then amount to the value of the emissions injection only. Due to the pulse response, the temperature response will depend on the time point we are in. Following the pulse response evolution, we will detect a minimal magnitude of temperature in the first year of the pulse and the maximal temperature at the peak of the response, ∼17 years after the pulse. Effectively, we have two very different temperatures for the

same cumulative emissions. The difference between those two temperatures gives the maximally possible scenario-dependent carbon budget deviation. If the cumulative emissions then amount to 100 GtC, the pulse response scales accordingly, and the theoretical deviation would be ~0.17 °C between the minimal and maximal response. However, since we introduced the slope restriction in the optimization process and the initial emissions in 2020 counted around 10 GtC, the emission pathway is not nearly as steep, resulting in smaller maximal deviations. In essence, the pulse response shows that if we want to maximize the

temperature response in a given year, we stack the emissions ∼17 years before that year; conversely, if we want to minimize the temperature response, we stack the emissions as close as possible (dictated by $k$) to that year.

       Finally, because of the relaxation of the response that comes in time, if we are far enough from the time where we maximized the deviation, the deviation itself diminishes – as shown in Figure 2. In the extreme case presented in the previous paragraph, that can be intuitively seen as follows. Albeit we could have a considerable difference in temperature stemming from the same

cumulative emissions between the 0th (the injection year) and 17th year (the peak year) following the pulse, if we go further in time, the temperature response difference between the 80th and 63rd year following the pulse (likewise, 17 years difference) is almost non-existent. Hence, the carbon budget deviation "fixes" itself as the system enters dynamic equilibrium, which is an analogue of the pulse response reaching a nearly constant value (relaxation).

       With this intuitive reasoning behind the (small) scenario-dependent deviations, we deduct that the pulse response shape is

critical for the climate model's adherence to the carbon budget approach. The FAIR model shows small, scenario-dependent deviations precisely because its pulse reaches an almost constant regime relatively quickly following a peak. If a model cannot emulate reaching the temperature relaxation, it will also show much higher emission scenario-dependent deviations.

### 4.2    Pulse Response Shape's Alteration as a State Dependency Indicator

Until now, we have utilized (Sect. 3) and inspected (Sect. 4.1) a singular, non-changing pulse response. However, the exper-

iment shows that this pulse response is not constant with the climatic conditions: following the same procedure described in 2.2.2, we generate pulse responses later in the RCP emission run, accordingly. The generated pulses are depicted in different colours in Fig. 4.

       When comparing the pulses, a general trend can be detected. As the system receives higher climatic stress in the form of higher cumulative emissions and higher temperatures, the pulse response changes. While all the pulse response variations show

the aforementioned steep increase in the first few years following the pulse, the magnitude of the peak and the corresponding relaxation response decrease with changing climatic conditions.

       This allows us to explain the detected widening gap between the full-fledged and Green's model's generated maximal and minimal temperatures that increase with higher $F_{\mathrm{tot}}$. Green's model uses a non-state dependent (non-changing) pulse response





as Green's function (Green1). Therefore, it shows higher temperature anomalies for both maximizer and minimizer compared
with the full-fledged model, which by its construction, is state dependent in every sequential timestep (see Appendix). The
difference between the two models gets more significant the more stressed the climatic system is, as shown in Fig. 3, left
panels.

Additionally, to the widening gap between the two models' generated maximal and minimal temperatures, we have detected
the widening gap between the corresponding carbon budget deviations $T_\mathrm{d}(k)$. As we can detect in Fig. 3, right panels, the gap
increases in favour of the lower full-fledged model's acquired $T_\mathrm{d}$. This can be explained by the flattening of the pulse response
curve with higher climatic stress. Green's approach utilizes the Green1 pulse response, which has a distinctly higher "peak
belly" than its counterparts. Conversely, the full-fledged model incorporates feedback, and its response acts accordingly, as
these two effects show.

One could have opted for a different Green's function that would reduce the aforementioned gaps between the full-fledged
and Green's model, but then the choice is case specific. Green's model would be more precise depending on if we generated
Green's function closer to the climatic conditions we want to inspect. Hence, Green's Approach can indeed be seen as a
linearized and simplified full model. Hypothetically, if the change of the pulse response $f_g$ with climatic conditions could be
incorporated into Eq. (2), the gap between the two approaches would decrease – if not become non-existent.

## 5 State-Dependent Deviation

As discussed in the introduction, the previous literature has suggested that TCRE is not a constant value but slowly decreases
with cumulative emissions. This can be interpreted as the carbon budget (climate) state dependence. While it was previously
numerically detected, here we offer a way to quantify it explicitly in the form of a new, state-dependent carbon budget equation.
The changing temperature response pulse is crucial in deriving the new equation.

### 5.1 State-Dependent Pulse Response Approximation: a State-Dependent TCRE

In Subsect. 2.2.3, it was already shown how the "perfect budget" pulse response in Green's model translates into the TCRE
included in Eq (1). If the TCRE decreases with background conditions, the "perfect" step pulse should also decrease in magni-
tude following the climatic stress. Indeed, we showed how that actual pulse response decreases in magnitude with background
conditions. If we approximate it with a step function, the decrease of the pulse response can be directly linked with the decrease
of TCRE. Keep in mind, however, that with that approximation, we lose the ability to express the time delay and scenario de-
pendency; the shape of the pulse response function dictates the scenario dependency. Motivated by the findings in Subsect. 4.2,
we attempt to explicitly quantify the TCRE dependency on climatic conditions in this subsection. The method employed is as
follows.

To generalize the analysis, we generate pulses under RCP4.5 and RCP8.5 emission scenarios, in addition to already generated
pulse responses under different climatic conditions with RCP6. We mimic the procedure from the last subsection and the
method section, generating the first pulse at the benchmark year of 2020 and the rest at the same temperature levels, where





possible. For RCP6, the climatic conditions under which the comparison pulses were generated are listed in Fig. 4 caption. Using RCP4.5 scenario, pulses are generated in years $t_{\text{RCP45}} = (2020, 2051, 2100)$ yr and the corresponding climatic conditions $F_{\text{RCP45}} = (588, 944, 1259)$ GtC, $T_{\text{RCP45}} = (0.96, 1.5, 1.94)$ °C and $C_{\text{RCP45}} = (404, 482, 529)$ ppm. For RCP8.5, the values are $t_{\text{RCP85}} = (2020, 2043, 2061, 2075)$ yr, $F_{\text{RCP85}} = (614, 966, 1354, 1716)$ GtC and $T_{\text{RCP85}} = (0.98, 1.5, 2.03, 2.51)$ °C.

Next, recalling the "perfect budget" discussion, we approximate the generated pulses with the step function. Ignoring the temperature evolution dynamics in the early years of the pulse response, the pulse is approximated with a constant $\Lambda$ by averaging it between years 70 and 80[7]. As shown in Fig. 4, the pulse dynamics relaxed by that time, reaching a relative constancy. Using this approximation, we ignore the temperature response dynamics. We justify this choice by the fact that extreme scenario-dependent deviations generated are relatively small and only temporary. In the long term, the constancy of 455   the pulse dominates the behaviour. After approximating the pulses, to each value of generated $\Lambda$ we assign its corresponding cumulative emissions and temperature values (e.g., the background climatic conditions under which the original pulse was generated). With that, we have a mapped $\Lambda(T, F)$ dependency, which when reasoned in line with Eq. (1)[8], can be considered a TCRE dependent on cumulative emissions and temperature increase.

With the first effect, the carbon budget state dependency is made explicit. Looking at each RCP case separately, we can infer 460   that $\Lambda$ decreases linearly both in $T$ and $F$. Moreover, looking at the right figure, we can see that by adding 1000 GtC, $\Lambda(F)$ dropped by roughly 10%, which is in accordance with the findings from Leach et al. (2021).

Secondly, the linear relationship between $\Lambda$ and $T$ and $F$ seems to change depending on the RCP scenario in which it was generated. We can see that the (negative) slope of $\Lambda(T)$ and $\Lambda(F)$ increases as the emission scenario becomes more emission intense. In absolute terms, it is lowest for RCP4.5 and highest for RCP8.5. This finding indicates that there could be another 465   layer of carbon budget deviation, which could be seen as a "scenario-dependent state dependency". We take that finding with a grain of salt since it comes about following several approximations – investigating the underlying reasons is left for future research.

## 5.2   State Dependent Carbon Budget Equation

If we focus only on one RCP runs' generated $\Lambda$'s, we can analytically derive the carbon budget state dependency. For RCP6 scenario, we acquire $a$ and $b$ by linear approximation of the point values $\Lambda(T)$, which gives $a = 1.083 \cdot 10^{-4}$ GtC$^{-1}$ and $b = 1.646 \cdot 10^{-3}$ °C GtC$^{-1}$, so that $\Lambda(T) = -a \cdot T + b$.

Since $\Lambda$ is, by definition, a temperature response to an emission pulse, we interpret the temperature change following the approximated pulse as $\Delta T = \Lambda(T) \cdot E_{pulse}$. In other words, the temperature change is equal to one unit pulse emission scaled 475   by temperature response to a pulse $\Lambda$. Using the fact that the emission pulse brings about the change in cumulative emissions

---

[7]In that way, the approximation for each pulse resembles the black dashed line in relationship to the blue line in Fig. 4.

[8]Note that $\Lambda$ and $\lambda$ have the same function in the "perfect budget" equation. The difference is that $\lambda$ is a constant, while $\Lambda$ is a function of temperature and cumulative emissions.



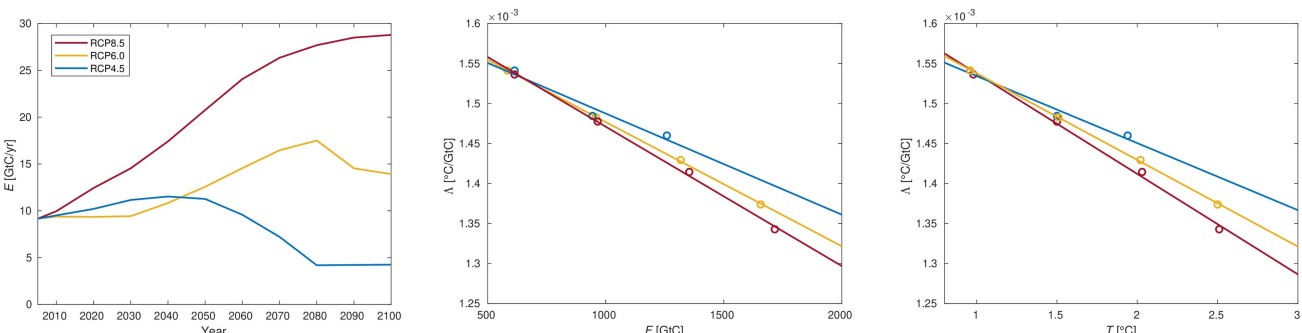

**Figure 5.** TCRE approximations Λ generated from pulse response functions under different climatic conditions and emissions scenarios. The middle panel shows Λ's dependency on cumulative emissions, and the right panel on global mean temperature anomaly. Scatter plots are actual values of Λ, while lines are corresponding linear extrapolations. Different colours represent Λ dependencies generated from different RCP scenarios that are plotted in the left panel.

we rewrite the forementioned relation in the form of a differential equation:

$$\mathrm{d}T = (-\mathrm{a} \cdot T + \mathrm{b})\mathrm{d}F. \tag{5}$$

Solving this form of differential equation analytically is straightforward. Hence, integrate Eq. (5), and we get:

$$T(F) = \frac{b}{a} + (T_0 - \frac{b}{a})e^{-a(F-F_0)}, \tag{6}$$

In its essence, Eq. (6) represents an improved, state-dependent version of the carbon budget equation. Instead of linear, it has a weakly negative exponential temperature dependence on cumulative emissions.

To get information about its shape in the relevant domain, we insert the values of the constants. Since we generated pulse responses only from 2020 onwards, we chose that year for our initial conditions[9], so $T_0 = 0.955$ °C and $F_0 = 584$ GtC. The values of a and b are listed in a few paragraphs above. Numerically, we get $T(F) = 15.1985 - 14.2434e^{-0.0001083(F-583.58)}$.

If plotted, one can see that $T(F)$ is a closely linear, slightly concave function within the $F$ domain of interest[10]. What is important is that the ratio between temperature and cumulative emissions changes with changing cumulative emissions, and hence, it captures state dependency.

As a side note, one could ask why we derive the equation with Λ mapped with temperature instead of cumulative emissions, as both pieces of informations are available in Fig. 5. The main argument is that we deem temperature to have more physical

meaning than cumulative emissions when it comes to the natural system. Nevertheless, we could easily redo the derivation with a mapped $\Lambda(F)$ and get another form of state-dependent carbon budget equation where the temperature is a quadratic function

---

[9]These are the same initial conditions as for the full-fledged approach in the optimization run.

[10]Note that $F$ here represents the total cumulative emissions from the preindustrial era. One could rewrite the equation with $\Delta F = F - F_0$ to get the temperature increase relative to the initial year, here chosen to be 2020.





of cumulative emissions. If a user finds quadratic form more useful[11], it is not a problem since the two equations give the same temperature diagnoses, at least in the tests done for Eq. 6.

To check if the newly introduced Eq. (6) is reasonable in that it generates correct temperature dynamics, we test it with RCP
emission scenarios plotted in Fig. 5, left panel. We plot the generated temperature evolutions in Fig. 6 (red lines) alongside the full-fledged FAIR output (blue lines) and the carbon budget Eq. (1) with constant TCRE (yellow lines) for each RCP scenario. Firstly, however, we scrutinize the implication of choosing a constant TCRE.

To understand how the choice of constant TCRE affects the temperature evolutions, we use two TCRE values: the higher one, $TCRE_{v1}=1.6 \cdot 10^{-6}$ °C GtC$^{-1}$, and the lower $TCRE_{v1}=1.53 \cdot 10^{-6}$ °C GtC$^{-1}$, the latter being given as a mean TCRE
value in Leach et al. (2021). As we can see in the figure, the higher choice of constant TCRE results in an accurate temperature diagnosis in the first half of the century. However, it overestimates the temperature in later years (higher cumulative emissions) compared to full-fledged FAIR. The opposite is true for the lower choice of TCRE, and we can discern that in Leach et al., TCRE was gauged to fit higher climatic conditions. In that sense, Eq. (1) with a constant TCRE is a linearized version of FAIR in a similar way as Green's function model but without the ability to generate scenario dependent effects. Additionally, we can
detect that the state-dependent deviations are not transient as their scenario-dependent counterparts, but ever-increasing with the changing cumulative emissions. The highest detected deviation is around $\sim 0.5$ °C for the end-of-the-century temperatures in the RCP8.5 run.

Unlike constant TCRE, our newly introduced state-dependent carbon budget Eq. (6) replicates full-fledged generated temperatures in RCP4.5 and RCP6 runs almost perfectly. A drift is detectable for higher temperatures in the RCP8.5 and can be
attributed to the $\Lambda$ having a steeper linear decrease with $F$ and $T$ (Fig. 5) under the RCP8.5 scenario than the RCP6 one, the latter being chosen to generate Eq. (6). The maximal drift from the full-fledged model's generated temperature is around 0.1 °C at the end of the century. Since RCP8.5 is arguably somewhere in the upper boundary for possible emission pathways, we can safely say that Eq. (6) is a good emulator of the full-fledged FAIR model and a state-dependent enhanced carbon budget equation, at least for the standard parametrization used throughout the paper. The incorporation of climate parameters in Eq.
(6), and its scrutinization in view of climate economics will follow this work.

## 6   Discussion

In this paper, we focused on the deviations from the carbon budget, formalizing a distinction between the carbon budget emission scenario-dependent and climate state-dependent deviation, both logically independent. By state-dependent deviation,
we mean the change of TCRE value, depending on the background climatic conditions – specifically, the cumulative emissions and global mean temperature increase. By scenario-dependent deviation, we mean the possible difference in temperature that comes solely from the preceding emission choice while keeping cumulative emissions fixed. We explicitly calculated the maximal possible scenario-dependent carbon budget deviations. Furthermore, we offered a reinterpretation of the carbon budget

---

[11]The quadratic form reads as: $T(F) = -\frac{a'F^2}{2} + b'F + (T_0 - \frac{a'F_0^2}{2} - b' \cdot F_0)$

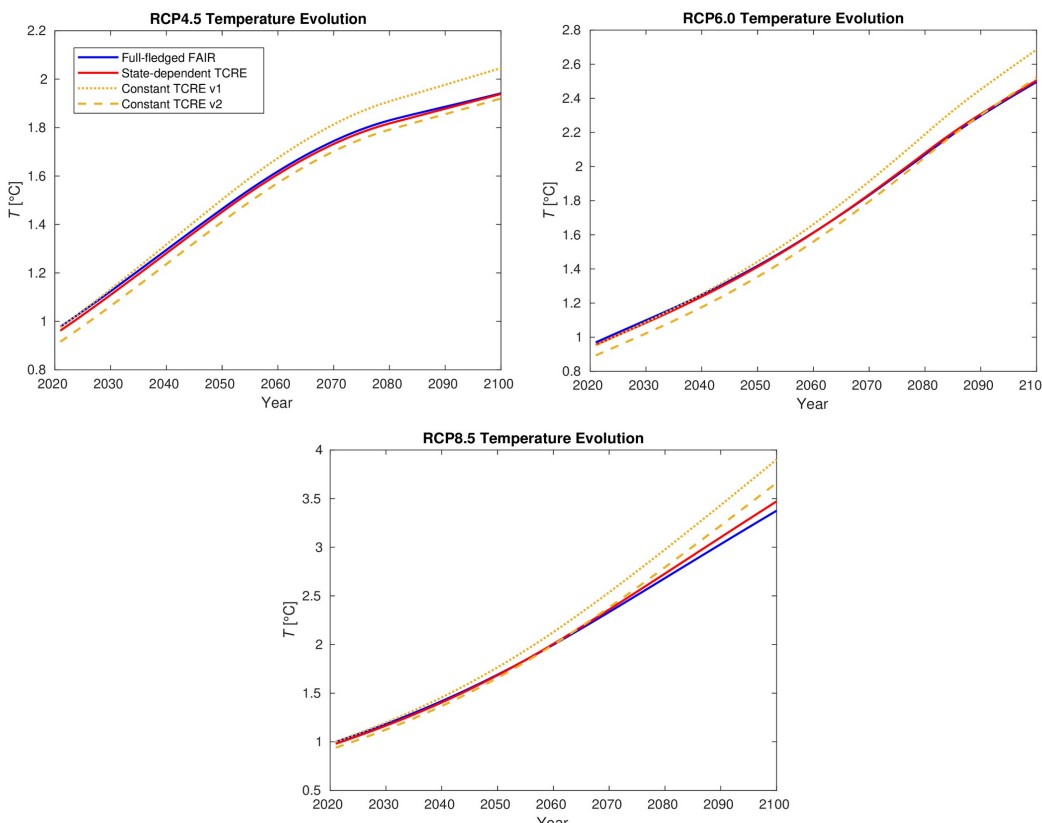

**Figure 6.** Temperature evolution under the three RCP emission scenarios, as calculated by the full-fledged FAIR model (blue), the newly introduced state-dependent carbon budget equation (Eq. (6)) (red), and the "perfect budget" equation (Eq. (1)) with two different TCRE values (yellow). The state-dependent carbon budget equation is a very good emulation of the full-fledged model. In contrast, the constant carbon budget equation can emulate the full model in a specific domain only, depending on how it is calibrated; e.g. for a higher TCRE chosen value (v1), Eq. 1 is a good fit in the early years of the RCP runs but starts to drift away from the full-fledged model with higher temperatures.





approach through the lens of Green's function formalism, encapsulating the scenario dependency. We connected the change of
pulse response under climatic change and derived a new carbon budget equation; henceforth, we derived a new analytical form
of the carbon budget equation that captures state dependency. This last section summarises the findings and emphasises their
broader implications.

Section 2 introduced the FAIR model, a simple climate model with four carbon and three temperature compartments. The
FAIR model is chosen for the analysis for several reasons: firstly, it has the ability to capture climate feedback on the carbon
cycle; secondly, the model has already been praised in the literature for its efficiency; thirdly, it is relatively easy to implement
and is computationally cheap; lastly, and most importantly for this paper, it can correctly capture the temperature response to
emission pulse (e.g., pulse response). As we show in the paper and deem it the central finding, the pulse response behaviour
reveals the climate model's degree of both scenario and state-dependent carbon budget deviations.

To test whether pulse response behaviour is trustworthy to give us information on carbon budget deviations, we incorpo-
rated it in the form of Green's function. We have shown that if the relationship between temperature increase and cumulative
emissions is purely linear without any time delay, the pulse response ought to be a constant step function, independent of
background climate conditions. Plugging in the step function as a Green's function indeed leads to the "perfect carbon budget"
equation which gives the first proof of Green's concept under the umbrella of the carbon budget approach.

In Sect. 3, we compare the pulse response (Green's) concept with the full-fledged model under the optimization procedure
that maximizes/minimizes the temperature in a run under fixed cumulative emissions. In this experiment, we effectively test
both the extreme cases of scenario-dependent deviations (focusing on strictly FAIR model results) and numerically test whether
the introduced Green's approach can replicate the full-fledged model.

When quantifying the scenario-dependent deviations, it was shown that how we emit after the optimization year would
drastically affect the generated deviations; we differ two cases: the "net-zero" and the "open budget" interpretation. In the
former, no emissions are allowed after the optimization year $t^*$, de facto reaching and staying at net zero after $t^*$; in the latter,
that condition is non-existent. The highest possible deviation we acquire is approximately 0.15 °C for the "open budget" case.
In the net-zero case, the highest deviation is way below 0.1 °C. From the policy-relevant carbon budget viewpoint, this is good
news as it keeps the carbon budget approach resistant to scenario choice when complying with specific temperature targets and
net-zero commitments. Regardless of the interpretation, the carbon budget scenario-dependent deviations we detected are not
permanent but a result of the optimization in one year. The arguably small deviation subsides relatively quickly if we allow
the system to go towards dynamic equilibrium. Furthermore, we showed that scenario-dependent deviations increase with the
higher cumulative emissions cap but do not dependent on the year where we maximize the deviation. Finally, it is important to
emphasize that the acquired scenario-dependent deviations are the extreme cases, not the plausible future realities. We conclude
that the carbon budget approach is robust to scenario choice.
The second result of Sect. 3 is that the pulse response in the role of Green's function can indeed replicate the full-fledged
generated scenario-dependent deviations to a great extent. Combined with the formal connection between the "perfect budget"
and Green's approach, this implies that the pulse response shape can be used to indicate the source of the model's associated
scenario-dependent deviations. The conclusion is that the FAIR model shows relatively small scenario-dependent deviations





because its pulse response reaches a close constant relatively fast as a result of its combination of temperature and carbon

boxes' dynamics. In essence, we deduct that the pulse response's shape dictates scenario-dependent carbon budget deviations. Following the shape comparison of pulse responses presented by Fig. 1 in Dietz et al. (2021), we conclude that most simple climate models have some potential for carbon budget scenario dependency because of their inability to reach a constant temperature response – adding to the argument for replacing climate models in integrated assessments with FAIR. Dietz et al. (2021) argued that FAIR should be used in climate economics, referring to its pulse response representation being able to

emulate the complex models' pulse response to justify it. Here, we connected the pulse response representation with the roots of the carbon budget and its deviations.

Another finding comparing the optimization runs is an ever-increasing gap between the two approaches' generated temperatures with the increasing cumulative emissions. This effect occurs because we are restricted to using only one pulse response as Green's function. Therefore, it cannot capture the state dependency as the full model with its non-linear processes. Nev-

ertheless, the state-dependent effect can be detected when different pulse responses are generated under different climatic conditions. We have shown that the pulse response decreases in magnitude and flattens with higher cumulative emissions and temperatures. This finding in combination with a relatively flat shape of the pulse response function is used to generate a new, state-dependent carbon budget equation. The procedure is briefly reiterated as follows.

Since we have shown that FAIR-generated scenario dependencies are small and non-permanent, we approximate each of

the (state-dependent) pulse responses with a step function. Hence, we return to the "perfect budget" Green's function case, but with a pulse response step function that changes with climatic conditions, de facto imitating a change of TCRE with every additional emission pulse. Finally, with the methodology explained in detail in Sect. 5, we derive a state-dependent carbon budget equation. Tested against the full-fledged model under the RCP scenarios, the new equation imitates the model almost perfectly for temperatures below 2.5 °C, when it starts to drift as a constant TCRE counterpart. The persistent, state-dependent

deviation at the end of the century of RCP8.5 run is ~0.5 °C when using a constant TCRE, and ~0.1 °C when utilizing a state-dependent carbon budget equation. Since the full-fledged model shows the temperature anomaly of 3.4°C, the maximally achieved state-dependent deviation drops from around 30% to roughly 3% when switching from the constant (Eq. 1) to a state-dependent TCRE (Eq. 6). Additionally, the new proposed carbon budget equation has an exponential form, which makes it very convenient for analytical assessments due to its functional properties. With its new, state-dependent form, we believe

there is potential for the utilization of carbon budget approach in climate economics analytical assessments. Furthermore, what is genuinely surprising is that we have managed to capture the whole model with seven compartments into a single equation without losing too much precision.

In essence, this paper offers two ways to reformulate the carbon budget approach equation – Green's function formalism and the state-dependent carbon budget equation. One can recreate scenario-dependent effects, the other state-dependent drop

of TCRE. The scenario-dependent effects are small, maximally around 7% of overall temperature change. Moreover, they are not constant in time due to the shape of the pulse response. On the other hand, state-dependent deviations are permanent, ever-increasing as a function of cumulative emissions, with a possible contribution to a drift of 30% from the full-fledged





model's generated temperatures. Hence, tackling the state-dependency makes the state-dependent equation a likely candidate to substitute the full-fledged model in view of the carbon budget approach.

As an outlook for the follow-up research, we plan to thoroughly test the viability of the state-dependent equation. Firstly, the analysis ought to be redone so that the derived equation also includes the climate sensitivity or transient climate response parameter. This is crucial for sensitivity analyses which are essential in integrated assessments. Secondly, the equation is to be tested in some integrated assessment model which takes carbon only into account (e.g., MIND (Edenhofer et al., 2005)), and see how the objective function (e.g. welfare) changes if we replace the full climate model with one equation that does not
include time-delay.

**Conclusion**

The current view of the carbon budget approach, if the aspiration is for it to be utilized in economic assessments, suffers from the inability to capture the emission scenario and climate state dependency. The latter causes more significant issues than the former, as it causes a permanent deviation from the assumed budget under the cumulative emissions. Additionally, cumulative
emissions is a monotonically increasing function, and so is a state-dependent deviation. Conversely, the scenario dependency will primarily be affected by the model used or, more precisely, by the pulse response shape of the model used. In this paper, we reaffirmed the value of the FAIR climate model due to its simplicity and pulse response shape, which results in relatively small scenario-dependent deviations. Using that fact, we generated different pulse responses under different climatic conditions and approximated them with a state-dependent TCRE, which led to a fresh reformulation of the carbon budget approach.

**Appendix A:  FAIR model**

The Finite Amplitude Impulse-Response (FAIR) model was first introduced by Millar et al. (2017). In essence, it consists of the climate state-dependent modification to the carbon cycle of the simple climate-carbon cycle model proposed in Joos et al. (2013), the latter of which does not include climate feedback. Following the FAIR introduction that consisted only of the carbon cycle and the associated radiative forcing, Smith et al. (2018) added the dynamics of other (non-$CO_2$) radiative
forcers, hence introducing FAIRv1.3. The newest (to our knowledge) version of FAIR is published (FAIRv2.0.0) by Leach et al. (2021). It incorporates the non-$CO_2$ dynamics from FAIRv1.3, but differs from the original in the way it represents the carbon feedback parameter and in the number of temperature boxes. Initially, FAIRv2.0.0 was introduced because of the more widely approachable feedback representation, which simplifies the pogramming scheme used for its employment. In GAMS, two representations are equivalent in programming difficulty level (as opposed to in, e.g. python). Nevertheless, in this paper,
we use v2.0.0 as it is the most up-to-date FAIR version.

The FAIR model can be roughly divided into two modules, the carbon cycle module and the radiative forcing (temperature) module. Here, we provide a more detailed description of the model than in the main text, alongside a full set of equations and



a complete utilized parameter values list. The following equations constitute the carbon cycle module:

$$\frac{\mathrm{d}R_i(t)}{\mathrm{d}t} = a_i E(t) - \frac{R_i(t)}{\alpha(t)\tau_i}, \, i = 1, 2, 3, 4, \tag{A1}$$

$$C(t) = C_0 + \sum_{i=1}^{4} R_i(t), \tag{A2}$$

$$\alpha(t) = g_0 \cdot exp\left(\frac{r_0 + r_u G_u(t) + r_T T(t)}{g_1}\right), \tag{A3}$$

$$G_u(t) = \sum_{s=t_0}^{t} E(s) - \sum_{i=1}^{4} R_i(t) \tag{A4}$$

where, $g_1 = \sum_{i=1}^{4} a_i \tau_i \left[1 - (1 + 100/\tau_i)\,\mathrm{e}^{-100\mathrm{yr}/\tau_i}\right]$ and $g_0 = \exp\left(-\frac{\sum_{i=1}^{4} a_i \tau_i \left[1 - \mathrm{e}^{-100\mathrm{yr}/\tau_i}\right]}{g_1}\right)$. Four carbon compartments $R_i$ constitute the carbon cycle, with each compartment being empty ($R_i = 0$) in preindustrial conditions. The total sum of the carbon content in the compartments in addition to the preindustrial concentration $C_0$ gives us the atmospheric carbon concentration $C(t)$ (Eq. (A2)). The carbon dynamics of the compartments $R_i$ are given in Eq. (A1). The $a_i$ fraction of emitted carbon $E(t)$ enters the $i$th compartment[12], while the content of the compartment decays with the rate of $\tau_i$. The decay timescale is adjusted by the scaling factor $\alpha(t)$, which introduces the feedback mechanism in this model and hence effectively, the decay timescale at each timestep is given as a product of two ($\alpha(t)\tau_i$). Equation (A3) shows $\alpha(t)$ dependence on the global mean temperature $T(t)$ and the cumulative emissions taken up by the carbon sinks[13] $G_u(t)$ (Eq. (A4)). The intuition why we have such a dependence is given in the main text. Finally, from the carbon cycle module the atmospheric concentration $C(t)$ is extracted and translated into radiative forcing.

The radiative forcing part and the sequential warming dynamics are described with the following set of equations:

$$F(t) = f_1 \ln \frac{C(t)}{C_0} + f_2 \left[\sqrt{C(t)} - \sqrt{C_0}\right], \tag{A5}$$

$$\frac{dS_j(t)}{dt} = \frac{q_j F(t) - S_j(t)}{d_j}, \, j = 1, 2, 3, \tag{A6}$$

$$T(t) = \sum_{j=1}^{3} S_j(t). \tag{A7}$$

---

[12]Nota bene, the emissions are usually given in units of GtC, while $R_i$ is in ppm. Hence, $E(t)$ is divided by the conversion factor 2.12 in Eq. (A1). Conversely, $G_u(t)$ is counted in GtC

[13]Note that in the original form given by Leach et al, there is one added dependency on the gas atmospheric burden $G_a(t)$. However, if the analysis of methane is excluded in standard parametrization, the $\alpha(t)$ dependency on $G_a(t)$ equals zero.



First, the atmospheric concentration gives rise to the radiative forcing $F(t)$ through a combination of a logarithmic and square root term (Eq. (A5)). Finally, the forcing increases temperature as given by Eqs. A6 and A7. Following the similar structure

as the carbon cycle, Equation A6 gives the temperature response of each box to radiative forcing, with the corresponding timescales $d_j$ and parameters $q_j$ that define the equilibrium temperature response. Finally, the global mean temperature anomaly $T(t)$ is calculated as the sum of three components in Eq. A7.

The first version of FAIR incorporates two temperature boxes, following the example set up by Geoffroy et al. (2013). Following the recent literature that suggests three box model is more appropriate to capture the behaviour observed in the

state-of-the-art (CMIP6) climate models (Tsutsui (2020), Cummins et al. (2020)), Leach et al. (2021) introduce an additional (third) box to the FAIR temperature module. It might be worth noting that when we say compartments in this context we actually mean the modes of the compartments system after the diagonalization of the matrices that represent the actual carbon and temperature reservoirs and their correlated dynamic flows. This is why both atmospheric content and the temperature increase are the sum of the carbon and temperature "compartments". We stick to the nomenclature and do not name them

modes in this work in order to follow the previous literature tradition.

The standard parameter values are as follows: the emission uptake fractions of each carbon reservoir $(a_1, a_2, a_3, a_4) = (0.217, 0.224, 0.282, 0.276)$; atmospheric decay timescales $(\tau_1, \tau_2, \tau_3, \tau_4) = (10^6, 394, 36.5, 4.3)$ yr; feedback parameters $(r_0, r_u, r_T) = (33.9, 0.0188\,\mathrm{GtC}^{-1}, 2.67\,^\circ\mathrm{C}^{-1})$; forcing coefficients $(f_1, f_2) = (4.57\,\mathrm{Wm}^{-2}, 0.086\,\mathrm{Wm}^{-2}\mathrm{GtC}^{-1/2})$, thermal response timescales $(d_1, d_2, d_3) = (0.903, 7.92, 355)$ yr; equilibrium responses $(q_1, q_2, q_3) = (0.18, 0.297, 0.386)\,^\circ\mathrm{CW}^{-1}\mathrm{m}^{-2}$. The final two sets of

parameters ($d_i$ and $q_i$) are tuned such that the model has a default value for equilibrium climate sensitivity equal to 3.24 K and transient climate response 1.79 K.

*Competing interests.* The author has declared that there are no competing interests.

*Acknowledgements.* M. Bekchanov provided a GAMS-coded version of FAIR from Bekchanov et al. (2022). B. Blanz helped fixing the occurring code errors. V. Brovkin pointed out the paper in which the shape of the pulse response function was revealed (Joos et al., 2013). H.

Held triggered this work, suggested inspecting the Green's function directly from emissions to temperature, and to use the minimization and maximization operations to generate the folder of scenario dependence. I would like to thank B. Blanz and H. Held for helpful discussions. Lastly, I would like to thank H. Held for carefully reading the manuscript. This research was funded by the Deutsche Forschungsgemeinschaft (DFG, German Research Foundation) under Germany's Excellence Strategy – EXC 2037 'CLICCS - Climate, Climatic Change, and Society' – Project Number: 390683824.





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
