# Peer review of "Carbon Budget Concept and its Deviation Through the Pulse Response Lens"

_Earth System Dynamics, 2023_

## Referee Comment (RC1)

Review of: Carbon Budget Concept, its Deviations and the Equation: Climate Economics Perspective

**Overall Assessment:**

The paper examines the deviations from linearity and path-dependence in the cumulative emissions of $CO_2$ – global temperature relationship, using Fair and an analytical approximation of Fair. The goal of the paper is to find a way incorporate these effects into the simple models of carbon budget used in economic analysis. While the paper is mathematically sound, there are substantial conceptual and scientific problems, which prevent publication in its present form. I suggest that the paper be rejected with an invitation to re-submit.

**General Comments:**

**(1)** The key conclusion of the paper is that the common approximation of the cumulative emissions of $CO_2$ – global temperature relationship should be modified in order to account for the reduction in TCRE at higher cumulative emissions. While this conclusion was reached earnestly, it is an error originating from the use of Fair. Simple and intermediate complexity climate models show a clear diminishing TCRE at higher cumulative emissions, full complexity Earth system model do not. This was clearly shown in Tokarska et al. 2016, and has not been contradicted by CMIP6 results. Full complexity Earth system model show the linear relationship holds to 5000 PgC, well beyond what is needed for economic models. Figure 3 from Tokarska et al. 2016 is copied below:

[Figure]

**Figure 3 | CO$_2$-attributable warming as a function of cumulative CO$_2$ emissions, and the resulting ratio of warming to emissions for CMIP5 ESMs and EMICs. a**, Simulated CO$_2$-attributable warming as a function of cumulative CO$_2$ emissions based on historical and RCP 8.5-Ext (solid) and 1PCTCO$_2$ simulations (dotted) from four CMIP5 models. **b**, The ratio of CO$_2$-attributable warming to cumulative emissions at 5 EgC emissions (TCRE$_{5EgC}$, top row) for these CMIP5 models, compared with TCRE for these models and other CMIP5 models (grey crosses, middle row)[10], and an observationally constrained estimate of TCRE (bottom row)[10]. **c,d**, The same as in **a,b** but for seven EMICs.

**(2)** The Green function used to approximate Fair is a 3$^{rd}$ order approximation, so rigour is needed to establish weather it can accurately capture path-independence. The Green function is approximating Fair, which is approximating full complexity Earth system models, which approximate the natural Earth system. MacDougall 2017 showed that path-independence in the cumulative emissions of CO$_2$ – global temperature relationship originates from ocean carbonate chemistry. Fair does not explicitly represent ocean carbonate chemistry and it is unclear from Leach et al. 2021 if Fair can even capture path-independence (given the results of the paper likely can but this needs to be explicitly shown).

To address this point the author can conduct a simple series of idealized experiments with Fair using different rates of emission. An example of such a set is shown in MacDougall 2017 Figure 1 (copied below):

[Figure]

**Figure 1.** Surface air temperature (SAT) anomaly versus cumulative emissions of $CO_2$ curves for:
(a) simulations with the UVic ESCM forced with $CO_2$ that is uncoupled from the model's biogeochemical
components (radiatively coupled $CO_2$ – R-$CO_2$). (b) Simulations with the UVic ESCM forced with fully coupled
$CO_2$. (c) Solutions for the ZD²OM not accounting for ocean carbonate chemistry or land carbon uptake.
(d) Solutions for the full ZD²OM. All models are forced with scenarios where atmospheric $CO_2$ concentration
changes at a constant rate. Note that the horizontal scale is different in the right and left columns as R-$CO_2$
emission have an airborne fraction of 1.

**(3)** The translation of Fair to GAMS needs to be validated. In an appendix show Fair
simulations from the Python and GAMS versions of the model to confirm there has been no
coding errors.

**(4)** The author does not seem to be familiar with the concept of Zero Emissions
Commitment (ZEC). In the climate sciences we now make a clear distinctions between
TCRE and ZEC with TCRE describing the relationship between cumulative emissions of $CO_2$
and temperature while emissions continue, and ZEC quantifying the warming that occurs
after emissions cease (See MacDougall et al. 2020 for recent model inter-comparison and
background). In the present work the author is mixing up transient path-dependence and
ZEC effects, which, in fairness, are related (Koven et al. 2023) but need to be distinguished
to avoid confusion.

**Specific Comments:**

**Line 24-25**: 'inauguration' and 'skyrocketed' do not make sense in the context of these
sentences. 'inclusion' and 'rose' would be more appropriate.

**Line 28:** Need working group for SPM

**Equation 1:** Lower-case lambda is a very poor choice of Greek letter. Little Lambda is always used for the climate feedback parameter in climate sciences. Capital Lambda has been used for TCRE in other papers, but you use that letter for another variable.

**Line 46:** TCRE is the Transient Climate Response to Cumulative $CO_2$ Emissions. The methanotropes get angry if 'carbon' is used instead of $CO_2$.

**Line 46 to 50:** Not quite right. The diminishing ocean heat uptake is also very important (see MacDougall & Friedlingstien 2015).

**Line 52:** This effect only exists in simple climate models and EMICs. Full complexity climate models remain linear to at least 5000 PgC.

**Line 60:** MacDougall 2017 suggested that the two effects might actually dependent, with models with low linearity having higher path-independence and vice versa.

**Line 69 to 74:** This is just wrong. If ZEC is 0 (the most likely value) the TCRE effect is essentially instantaneous. The lag from the carbon sinks (mostly the ocean) compensate exactly for the lag from ocean heat uptake. If ZEC is not 0 then small lags are expected.

**Line 91:** Calling Fair a 'full fledged' is bizarre. Fair is a simple climate model, barely above a climate emulator.

**Line 111:** 'prescribed' is a poor choice of words, since this term has a very precise meaning in climate sciences (basically a boundary condition).

**Line 129:** '$CO_2$' not 'carbon'. Elemental carbon (graphite) is not very soluble, nor is methane.

**Line 164:** No rationale is given for using RCP6.0. Why not one of the SSPs? Especially since the $CO_2$ concentration for year 2020 is way off in RCP6.0. The observed value was 414 ppm, 12ppm above the prescribed value. Since non-CO2 forcing is not used why not just use the historical record?

**Line 185:** IPCC range is 1 to 2.3 with a best estimate of 1.65 K/EgC. See Table 5.7 AR6 WG1.

**Section 3.1.3:** This is just ZEC.

**Section 3.1.4:** Weird mix of idealized and historical scenarios. Why not just use an idealized experiment?

**Figure 1:** In the legend second red max should probably be a min?

**Line 345 to 349:** Should show these pathways in a figure so readers can see if they are physical.

**Figure 2:** Should the units for the shading be GtC yr$^{-2}$?

**Line 360:** Typo here

**Figure 4:** This should be figure 1 since it is mentioned in text first. Also SI prefixed exist for a reason, just give the units in mK and get rid of the hard-to-see x10$^{-3}$ in the corner.

**Figure 5:** Again SI prefixes are your friend.

**References:**

Koven, C. D., B. M. Sanderson, and A. L. Swann, 2023: Much of zero emissions commitment occurs before reaching net zero emissions. Environmental Research Letters, 18 (1), 014017.

MacDougall, A. H., 2017: The oceanic origin of path-independent carbon budgets. Scientific Reports, 7, 10 373, doi:10.1038/s41598-017-10557-x.

MacDougall, A. H. and P. Friedlingstein, 2015: The origin and limits of the near proportionality between climate warming and cumulative $CO_2$ emissions. Journal of Climate, 28, 4217–4230, doi:10.1175/JCLI-D-12-00751.1.

MacDougall, A. H., et al., 2020: Is there warming in the pipeline? a multi-model analysis of the zero emissions commitment from $co_2$. Biogeosciences, 17, 2987–3016.

Tokarska, K. B., N. P. Gillett, A. J.Weaver, V. K. Arora, and M. Eby, 2016: The climate response to five trillion tonnes of carbon. Nature Climate Change, doi:DOI: 10.1038/NCLIMATE3036.

---

## Referee Comment (RC2)

**General comments**

Firstly, apologies for the delay in receiving my review.

In this paper, Avakumovic examines the impacts of assuming that there is a strictly linear relationship between cumulative emissions of CO2 and warming in climate economics. They show that this assumption can lead to errors in assumed warming. They continue on to derive a new equation that represents the link between cumulative emissions of CO2 and warming in a way that also includes state-dependence. They note that neither Green's formalism nor their new equation can capture scenario-dependence and leave a formulation that captures this dependence for future work.

I think the paper is interesting, identifies an issue for climate economics and then provides a way to solve it. I believe the author can revise it to make it a useful contribution to the literature. However, I am recommending major revisions because I think it currently misses key literature and is not presented in a way that will have maximum impact.

**Major concerns**

**Missing key literature**

The author claims that "While [state-dependence of the relationship between cumulative CO2 emissions and warming] was previously numerically detected, here we offer a way to quantify it explicitly in the form of a new, state-dependent carbon budget equation". This isn't actually true, Nicholls et al. (2020) derived an equation that provides a way to capture non-linearities between cumulative CO2 emissions and warming (i.e. non-linearities in the carbon budget). Arguably, the Nicholls et al. formulation improves on the formulation presented in equation 6 here in two key ways: a) it provides a clean connection with the TCR and TCRE parameters assessed by the IPCC (via its $T\_2x$ and $C\_2x$ parameters), something the author leaves for future work (see their comments in the discussion, "the analysis ought to be redone so that the derived equation also includes the climate sensitivity or transient climate response parameter") and b) it can represent dominance of both the temperature response saturation and carbon cycle weakening terms i.e. it can show a concave or convex relationship between T and cumulative CO2 emissions whereas the author's proposal is always concave.

Nicholls et al. also investigate deviations of the relationship between cumulative CO2 emissions and warming from linear. The author's claim, "...we assert that the extreme cases of maximally possible scenario-dependent carbon budget deviations are yet to be investigated and scrutinized", may be true because Nicholls et al. did not optimise to maximise deviations, but it is a much less broad claim than is currently presented in light of the work done by Nicholls et al.

Missing this paper (unfortunately) undermines the author's claims of complete novelty. Nonetheless, Nicholls et al. did not consider the application to climate

economics or include the Green's formalism discussion so I think there is still space for this paper to present something new. However, it will likely require a significant restructure to focus more on the new parts and either compare their proposal to the results and equation of Nicholls et al. or simply use the Nicholls et al. parameterisation as their translation tool (removing their equation 6).

**Writing**

The paper reads like it hasn't been proof-read for basic errors (typos, missing words etc.) or overall structure and fluency. This is unfortunate, because it makes the paper's key points much harder work to understand than they could be. I appreciate that the author has already put a lot of work into this paper, but such changes are relatively simple to make and significantly improve the experience of the reader. A thorough proof and re-write to remove jarring language is needed.

The other big problem with the writing is that it is extremely slow in some patches. This makes the experience of reading the paper drag, which is unfortunate because there are lots of interesting insights so the experience should rather be one of learning and excitement. There are two obvious things that contribute to the slowness of the writing (one easier to fix, the other harder). In general though, I would ask the authors to consider how to make the writing punchier by removing areas of repetition and areas where they circle around the point they are making rather than getting straight to it.

The easy fix is making the language more active. Describing is slow. If you instead change your text so focus on what the point is, then things are generally easier to read. This can be achieved by making the sentence focus on the point, pointing to evidence in brackets e.g. "Scenario-dependence in FaIR is around 0.1C (line description, Figure X)." As another example, "That implies that the pulse response introduced in the previous subsection should also be a constant function. In Fig. 4, it is plotted in a dashed black line: the temperature response to an emission pulse has an immediate jump following the emissions, and it does not change in time, as the "perfect budget" implies." could be re-written as "That implies that the pulse response introduced in the previous subsection should also be a constant function (dashed black line in Figure 4)." In this section, there is also quite a lot of text spent on showing that the perfect budget can be represented by a heaviside function, which is a fairly trivial observation and not worthy of so much text I don't think (move it to supplementary if you really want to keep it). In general, I would reconsider whether the more trivial observations would instead be better in a supplementary or simply removed (for example the discussion of different ways one can reduce emissions and still have the same cumulative emissions in section 3.1.1).

The harder fix is how much background is included in the paper. This is tricky because the author is (arguably) writing for two audiences: climate economists and climate scientists. The problem is that what is boring, trivial background

for one audience may be vital background for the other. As a result, it may be inevitable that some sections do feel slow to some parts of the audience. I'm not sure if there is a perfect solution to this, but I would invite the authors to reconsider which parts need to be in the main text. For example, as a climate scientist I suspect much of the discussion of FaIR's structure could be moved to the appendix (or even to a supplementary) because repeating the underlying paper's isn't needed. Another example is the intro, which includes a very slow historical account of the use of the carbon budget by the IPCC (yet still misses the use of the carbon budget in SR1.5, which was arguably a key point too). I don't think this historical account is really needed at the very start of the paper, and it could be significantly shortened in my opinion.

**Key insights get lost**

The above point is a shame because some really nice key points get lost. The analysis in Section 4.1 is very nice and provides a very nice insight into how FaIR (and perhaps the climate system) behave and what this means for approximations based on the simple linear carbon budget equation. However, this point can be lost because it does not stand out from everything else that is going on in the paper. I thought Section 4 was very nicely done and it should, arguably, be a much more key part of the paper. If other sections need to be shortened to achieve this, I don't think that would be a bad thing.

**Journal fit**

I hesitate to bring this up as I know how hard it is to find a journal that fits interdisciplinary papers like this one. However, the authors state their audience is climate economists. However, ESD is not a primarily climate economics journal so I do wonder whether they need to broaden their audience, re-write to fit ESD's primarily climate science audience or re-consider whether they are in the right spot. Discussion with the editor will likely clarify this (I may also have not well-understood the journal's intended audience).

**Minor concerns**

**Clarity of methods presentation**

The methods aren't always clearly presented. The references to the optimisation in the abstract are very hard to understand without having first read the paper. I would suggest removing discussion of the optimisation in the abstract as this is a methodological detail whereas the abstract here could simply focus on identifying the problem and the solution.

On this point, the open budget isn't clearly presented. A (supplementary) figure demonstrating the methodology would be very helpful. In particular, it wasn't clear to me whether you enforce that $E(t > t^*) = 0$ in the open budget cause or whether there can be emissions after $t^*$ (which would seem odd, as then the

cumulative emissions that the system sees are now changing from the specified limit up to $t^*$). Clarifying this may also help clarify the extent to which the author is blending the TCRE and ZEC concepts or not.

**Introduction**

The introduction is a bit slow and doesn't really lay out what the paper does clearly. Could it be re-formulated to a much simpler structure, e.g.

- the carbon budget concept is widely used
- for climate economics, a cheap tool that goes beyond simple linear would be very helpful to avoid some clear errors that come with the simple linear approximation
- climate economists can't always use simple climate models because of . . .
- here I derive such a tool/use the equation of Nicholls et al. to see what the implications are/do both

**Lack of exploration of climate uncertainty**

The paper explores scenario- and state-dependence of the relationship between cumulative CO2 emissions and warming. However, the paper only runs FaIR in a default configuration so doesn't explore uncertainty in the climate response at all. This seems to be a shame: given that FaIR is a cheap model, the author could likely repeat the analysis to capture at least some of this uncertainty. That could be nicely explained through your impulse response view as changing the shape (both in magnitude of response and time-evolution) of the pulse response, helping the reader understand these differing climate system responses in a single framework. Such analysis could also lead to a very nice 'final product', specifically a set of parameters that could be fed into an updated carbon budget equation (whether the author's equation 6 or equation 4 of Nicholls et al. or both). Different values of these parameters would represent the range of climate sensitivities and provide a simple look-up table for climate economists who wish to include an updated carbon equation in their own models and also explore the range of climate responses consistent with the IPCC's latest assessment.

I think addressing this point would also address reviewer 1's major concern as some of FaIR's parameter combinations will exhibit a much more linear response (although even reviewer 1's own figures arguably still show the weak non-linearity and Nicholls et al. discuss this conclusion in detail).

**Validation of model and code**

I note that there is no section on code availability or validation of the GAMS implementation of FaIR. Both of these seem pretty key if reviewers are going to be able to assess the validity and reproducibility of the results (particularly so given that a known software package of FaIR is not used).

**Claims about FaIR being 'best'**

There are many places in the paper where claims that FaIR is the best model for climate economics are made (whether by the author or quoting Dietz et al.). I was surprised by these for two reasons. Firstly, when I looked at Dietz et al. I could not find the conclusion that 'FaIR was best for climate economics' (if the authors can point me to this that would be great). Secondly, using Dietz et al. is an odd choice of reference given that the IPCC included an assessment of different simple climate models in their latest IPCC report. This assessment comes in Cross-Chapter Box 7.1 of AR6 WG1 and I would recommend the author reads that and then updates their assessment of simple climate models based on it. Alternately, the author could simply remove all assessment of which model is best because it isn't relevant to their paper. What matters for the author is whether the model they use can capture the scenario- and state-dependence they are interested in. Cross-Chapter Box 7.1 of IPCC AR6 WG1 shows that to be the case, as does the author's own analysis, so all that needs be said is "FaIR 2.0 is good enough for the analysis we are doing here and is easy to implement in GAMS, so we use it" and then move on.

On this point, I think Cross-Chapter Box 7.1 of AR6 WG1 shows that FaIR is 'good enough' for negative emissions at the level of analysis you're doing here. There are lots of uncertainties when it comes to negative emissions, but for the level of precision you're going for it would be fine to think about negative emissions too and just note that there is (like everything) uncertainty, but this provides a step forward for economic assessment that is better than nothing.

**Full-fledged**

The authors keep referring to FaIR as 'full-fledged' FaIR. Unless their is a 'half-fledged' version of FaIR floating around, the author can remove the phrase 'full-fledged' everywhere and simply refer to 'FaIR'. The phrase 'full-fledged' is almost damaging because it gives the impression that FaIR is on the level of an Earth System Model, which it most definitely isn't (FaIR is an emulator or simple climate model).

**Lower cumulative emissions level**

At the moment, the lowest cumulative emissions level (including history) is 1000 GtC. Would it be worth adding a lower level, more in line with 1.5C of warming in total (I assume adding such a level is fairly trivial and this provides an extra, arguably policy-relevant, level).

**Leftover temperature**

It would be good to see a plot of T_left. Is having to prescribe T_left a problem? It seems an odd extra exogenous input to require. Could you just run the Green's model over the historical period too? This may have the benefits of adding

historical data as a validation point and making it easier to understand exactly how the changing TCRE affects model behaviour?

**Claims about time evolution of T_d**

One of the claims made is that T_d declines to zero. However, I am not convinced that this is a general thing rather than just being a model feature of FaIR, which appears to always reaches the same equilibrium for the same cumulative emms? I don't think you need to expand your analysis to use other simple climate models (you are of course welcome to) but I think you need to be more careful with the conclusions you make about this time dependence as this may just be a model feature rather than a general property of the climate system.

In the same vein, the following comment at the end of the paper should be considered more carefully, "Furthermore, what is genuinely surprising is that we have managed to capture the whole model with seven compartments into a single equation without losing too much precision.". Given that the whole point of FaIR is centred on impulse response (the "IR" in the name stands for impulse response) it isn't that surprising that its sum behaviour (particularly its default response) can be captured reasonably well by a single well-tuned impulse function. I would caveat this statement more carefully and consider testing the fit under a scenario where emissions decrease sharply (currently you've only tested your simplified formula under 3 scenarios of increasing or stable emissions), acknowledge that you've only tested it with CO2 forcing (so you avoid any non-CO2, particularly aerosol-related, headaches) and consider testing the fit under different parameter values that might not be so amenable to being captured by a single impulse response function. The AR6 distribution of FaIR 1.6 could be (fairly simply) translated to FaIR 2 I believe (because their carbon cycle and temperature response are basically the same) if you want a sensible range of parameter values to explore. The probabilistic distribution of FaIR 2 from RCMIP Phase 2 would also work I suspect.

**Claims about impact of optimisation year**

Concluding that optimisation year doesn't matter from a sample size of two doesn't seem super robust to me. I think if you're going to make this claim, a more systematic exploration of the sensitivity of results to the chosen optimisation year is needed.

**Why can't you just use a simple climate model**

One thing that wasn't clear to me was why climate economists can't just use a simple climate model. Are there cases where even the simple climate model is too expensive to run or hard to implement? If yes, it would be great to have just one or two sentences in the introduction that help the reader understand why a new (or almost new) equation is needed and why existing solutions don't work.

**Figures**

Figure 6 is a nice illustration. Could you make a plot where the y-axis is relative deviation from FaIR (even as a supplementary plot)? This would help assess the fit more easily (a 0.1C deviation in a scenario with 4C of warming isn't the same as a 0.1C deviation in a scenario with 1.7C of warming).

**Technical corrections**

'We' should become 'I' throughout given you are a single author

The figures are not ordered in the order in which they appear in the text. This isn't necessarily a problem, but it is very unusual and quite distracting for the reader.

FAIR → FaIR throughout (most unfortunately, FAIR is actually a different model)

page 1, line 8: 'scenario choice under the fixed cumulative' → 'scenario choice under fixed cumulative'

page 2, line 38: footnote 2 should be brought into the main text. The text is misleading without this caveat being in the main text

page 3, line 76: 'the optimization program' → 'an optimization program'

page 3, line 84: 'Furthermore, Using' → 'Furthermore, using'

page 3, line 90: 'the state-dependent carbon budget equation' → 'a state-dependent carbon budget equation' or do you believe you have find the one and only way of doing this?

page 4, line 115: "correctly capture the temperature response following one carbon emission pulse", this seems to miss the point and forces you to define 'correct' (which you currently don't). Surely the point is that FaIR is used elsewhere and has an adequately complex temperature response to explore the scenario and state-dependence of interest (which it does). You don't need to claim 'correctness' (which is arguably impossible to validate).

page 6, line 154: 'To use Eq. (2), must opt', missing word

page 6, line 170: 'Thus, pulse' → 'Thus, the pulse'

page 6, line 172: missing bracket after 'Green1'

page 6, line 178: 'that that' → 'that the'

page 10, line 292: 'the associated historical cumulative emissions counting 584 GtC' → 'the associated historical cumulative emissions being 584 GtC'

page 10, line 296: 'Conversely, this is not a problem for the full-fledged model since that "leftover" response is fed into the initial conditions of the run.' Repetitive of previous paragraph, can be deleted

page 11, line 308: 'due excluding' → 'due to excluding'

page 12, line 350: 'from from' → 'from'

page 13, line 360: 'dependency.irstly', something wrong

page 18, line 470: "one RCP runs'" → "one RCP run's"

Figure 1 caption: 'Max(T) in S1-G represents the minimal' → 'Max(T) in S1-G represents the maximum'

Figure 5 caption: "linear extrapolations" → "linear regressions" or do I misunderstand what I am looking at?

Heading sections: The headings seem to have random capitalisation: "Pulse Response as a Deviation Source" looks super weird, is this the journal's style?

**References**

Nicholls et al. (2020): Z R J Nicholls et al 2020, Implications of non-linearities between cumulative CO2 emissions and CO2-induced warming for assessing the remaining carbon budget, Environ. Res. Lett. 15 074017. https://iopscience.iop.org/article/10.1088/1748-9326/ab83af

IPCC AR6 WG1 Chapter 7: Forster, P., T. Storelvmo, K. Armour, W. Collins, J.-L. Dufresne, D. Frame, D.J. Lunt, T. Mauritsen, M.D. Palmer, M. Watanabe, M. Wild, and H. Zhang, 2021: The Earth's Energy Budget, Climate Feedbacks, and Climate Sensitivity. In Climate Change 2021: The Physical Science Basis. Contribution of Working Group I to the Sixth Assessment Report of the Intergovernmental Panel on Climate Change [Masson-Delmotte, V., P. Zhai, A. Pirani, S.L. Connors, C. Péan, S. Berger, N. Caud, Y. Chen, L. Goldfarb, M.I. Gomis, M. Huang, K. Leitzell, E. Lonnoy, J.B.R. Matthews, T.K. Maycock, T. Waterfield, O. Yelekçi, R. Yu, and B. Zhou (eds.)]. Cambridge University Press, Cambridge, United Kingdom and New York, NY, USA, pp. 923–1054, doi:10.1017/9781009157896.009. https://www.ipcc.ch/report/ar6/wg1/downloads/report/IPCC_AR6_WGI_Chapter07.pdf

---

## Author Comment (AC1)

Dear Reviewer,

First of all, let me express my gratitude for the examination and assessment of the manuscript. The points given are very well received and I am happy to reflect on those in the following text. Some of the points I will gladly incorporate in the revised manuscript (in case be given the chance), while for some I think extra clarification is needed in order to provide context, and aid the reader in understanding the concepts provided in the manuscript. This way, I contest the prospect of paper rejection argued in the review. In this response, I first give a general context of the research and I reply to what I detected as the main criticisms of the manuscript afterwards.

As apparently, in part, the goal of the paper was misunderstood, at first I summarize the key innovations as I see them:

(1) Deviations from a linear relationship between cumulative carbon dioxide emissions and global mean temperature are separated into path dependencies and nonlinearities and hence discussed separately in a more explicit manner. From the perspective of climate economics, this is relevant, as the relation between different decision-analytic frameworks is governed only by the former effect. Hence, the accuracy of approximation must only be expressed through the former effect.

(2) When analyzing path dependency, I do not rely on a limited number of stylized scenarios but systematically determine the maximum possible deviations. For this, FAIR only serves as a demonstration case. In spite of its nonlinearity, the Greens function approach explains most of the found deviations.

(3) I present a formula as one way of dealing with nonlinearity. It might substitute models like FAIR in regimes where FAIR should not be operated – either for being too computationally costly (endogenous learning under uncertainty) or for exaggerating nonlinearity. For the latter, it should be calibrated from ESMs. Again, FAIR only serves as a demonstration case.

(4) I demonstrate that path dependency even further reduces over time by an order of magnitude after emissions have been stopped, forming a bridge between TCRE and ZEC.

Referring to the paragraph „Overall Assessment":
The first sentence is correctly stating that the paper examines the deviations from linearity and path-dependence in the cumulative $CO_2$ – global temperature relationship with the FAIR model, with a caveat that the analytical approximation is one of the results, not a predefined tool.
Going into the second sentence, the goal of the paper is not to find a way to incorporate these effects into simple climate models used in economic assessments.
The goal of this paper is as follows. Firstly, we distinguish two different sources of deviations from the carbon budget as mentioned in the intro sentence, the (1) emission scenario (or path) dependence and the (2) climate state dependence manifested as a deviation from the above-mentioned linearity; both sources of deviations are treated independently. Accordingly, the goal of the paper is to quantify both deviations using FAIR.

Regarding the FAIR model & the climate economics perspective:
The usage of FAIR was critiqued in the review for various reasons. While I agree that the FAIR model is a simplified version of reality and lacks many natural processes, one should be aware that in climate economics the compromise between the ability to properly emulate the climate conditions (more precisely the GMT) in response to emissions on the side, and the computational costs on the other. Namely, the optimization procedures done in economic assessments require around 1 000 to 100 000 runs to find an optimal result, so the model needs to be as simple as possible, otherwise, the run would be never finished. In that sense, even MAGICC (Meinshausen et al. 2011) is considered computationally expensive and

FAIR has been suggested as the best compromise (Dietz et al. 2021). I will come back to FAIR when addressing the specific comments later in this document.

Furthermore, to contextualize the research, the research was partially titled „Climate Economics Perspective". Hence, I do not claim that the findings here reflect the real state of climate (i.e. the missing carbonate chemistry) but the best estimate in view of climate economics. In the context of climate economics and simple climate models, I argue that by inspecting the temperature response following an emission pulse (in the paper used as Green's function), one can concur to what extent a simple climate model adheres to a path-independent carbon budget. This gives extra weight to the paper, in addition to the two main goals of the paper.

With this stated, I would like to move to the main point of the response, referring to the general points raised in the review; some of the specific points I am glad to leave for the revision phase.

The first general comment argues that the deviation from the linearity is only observed in simple climate models and EMICs, while at the same time, the CMIP5 models in Tokarska et al. 2016 (Figure 3) show the linearity. To this, I have two arguments.

In Tokarska et al. 2016, the CMIP5 models were forced with both 1% per year atmospheric concentration increase forcing (pure carbon dioxide emission forcing) and the extended RCP8.5 scenario (aerosols and other non-$CO_2$ gases). The review argues that Tokarska et al. 2016 clearly show the linearity between cumulative emissions and the temperature increase with solid lines. However, those are the results using RCP8.5 scenarios so they have other non-$CO_2$ effects included, whereas FAIR was forced only with $CO_2$ in my work. Additionally, it is questionable if we have a linear relationship in a solid line for BCC-CSM 1.1 (green) in the relevant domain (~2000 GtC). On the other hand, it is not clear from the figure if the relationship is linear (or non-concave) for the purely $CO_2$ experiment (dotted lines). To round up, I quote from the paper „*Figure 3a shows that the warming in the RCP 8.5-Ext simulations scaled by the ratio of CO2 to total forcing, for a given magnitude of cumulative emissions, is slightly higher than for the 1PCTCO2 simulations. One possible reason for this is the warming from non-CO2 greenhouse gases, which reduces the diagnosed cumulative emissions in the RCP 8.5-Ext simulations associated with the carbon-climate feedback.*" A similar conclusion can be made by looking at the first figure that I attach below, which I refer to in my second argument.
With that in mind, I come to my second argument.

Looking at the graph given in the summary for policymakers AR5 WG1 (IPCC, 2013) in the first figure below, one can detect a slightly concave relationship between cumulative emissions and temperature increase, for RCP scenarios and the stylized 1% per year concentration increase scenario, also conducted in CMIP5 models. The figure can be found below. I drew the green and white lines to visually emphasize the deviation from linearity. In my research, only $CO_2$ emissions are taken into account, so the grey plume in Figure SPM.10 is imperative in this case. I use the result from the old report since in the newest SPM WG1 (IPCC, 2021), the relationship between cumulative emissions and the temperature increase is shown only for SSP scenarios (Figure SPM 10.).

[Figure]

Cumulative total anthropogenic CO₂ emissions from 1870 (GtCO₂)

Cumulative total anthropogenic CO₂ emissions from 1870 (GtC)

**Figure SPM.10 |** Global mean surface temperature increase as a function of cumulative total global CO₂ emissions from various lines of evidence. Multi-model results from a hierarchy of climate-carbon cycle models for each RCP until 2100 are shown with coloured lines and decadal means (dots). Some decadal means are labeled for clarity (e.g., 2050 indicating the decade 2040–2049). Model results over the historical period (1860 to 2010) are indicated in black. The coloured plume illustrates the multi-model spread over the four RCP scenarios and fades with the decreasing number of available models in RCP8.5. The multi-model mean and range simulated by CMIP5 models, forced by a CO₂ increase of 1% per year (1% yr⁻¹ CO₂ simulations), is given by the thin black line and grey area. For a specific amount of cumulative CO₂ emissions, the 1% per year CO₂ simulations exhibit lower warming than those driven by RCPs, which include additional non-CO₂ forcings. Temperature values are given relative to the 1861–1880 base period, emissions relative to 1870. Decadal averages are connected by straight lines. For further technical details see the Technical Summary Supplementary Material. {Figure 12.45; TS TFE.8, Figure 1}

Global surface temperature increase since 1850–1900 (°C) as a function of cumulative CO₂ emissions (GtCO₂)

[Figure]

**Figure SPM.10 | Near-linear relationship between cumulative CO₂ emissions and the increase in global surface temperature**

**Top panel:** Historical data (thin black line) shows observed global surface temperature increase in °C since 1850–1900 as a function of historical cumulative carbon dioxide (CO₂) emissions in GtCO₂ from 1850 to 2019. The grey range with its central line shows a corresponding estimate of the historical human-caused surface warming (see Figure SPM.2). Coloured areas show the assessed *very likely* range of global surface temperature projections, and thick coloured central lines show the median estimate as a function of cumulative CO₂ emissions from 2020 until year 2050 for the set of illustrative scenarios (SSP1-1.9, SSP1-2.6, SSP2-4.5, SSP3-7.0, and SSP5-8.5; see Figure SPM.4). Projections use the cumulative CO₂ emissions of each respective scenario, and the projected global warming includes the contribution from all anthropogenic forcers. The relationship is illustrated over the domain of cumulative CO₂ emissions for which there is *high confidence* that the transient climate response to cumulative CO₂ emissions (TCRE) remains constant, and for the time period from 1850 to 2050 over which global CO₂ emissions remain net positive under all illustrative scenarios, as there is *limited evidence* supporting the quantitative application of TCRE to estimate temperature evolution under net negative CO₂ emissions.

**Bottom panel:** Historical and projected cumulative CO₂ emissions in GtCO₂ for the respective scenarios.

Additionally, the exponential relationship between cumulative emissions and temperature derived in the paper only slightly deviates from linearity in the cumulative emissions regime that is inspected. This was not shown in the paper not to overflow the reader with plots and information, but it can be tested with the values provided in the text.

Albeit the relationship is nearly linear, the small and persistent difference in temperature output can make a large difference in economic assessments.

Lastly, even if the FAIR model does produce a higher deviation from linearity than the ESMs, one of the points of deriving the new equation (Manuscript, Eq. 6) was to show that one can fairly well approximate a FAIR climate model with a single equation if the state-dependent TCRE is considered (derived from FAIR). This is a result independent of the linearity discussion.

On the second point, it was never argued that Green's function is an approximation of the real climate system, but rather an approximation of a model; in this case a FAIR model.

Furthermore, the point of the paper was not to qualitatively detect the source of deviation from the real-world (natural) system, but to quantify what are the maximally possible deviations given the FAIR model, at the same time testing whether FAIR (and its Green's approximation) can accurately capture path-independence. Hereby, I refer to the comments questioning both models' ability to capture the path independence. In my paper, the path dependence was tested by the optimization program given by Eq. 4. By using the optimization procedure, the full portfolio of all the possible emission pathways (under the given constraints) was tested, and not just „a simple series of idealized experiments using different rates of emissions", as suggested in the review. Hence, using the optimisation program the pathways that generate the minimal and maximal temperature in a given year (y*) under the same cumulative emissions are generated. By subtracting those two, the maximal path-dependent deviation of the carbon budget is generated. Seeing it is small in magnitude, the conclusion is that the FAIR adheres to scenario independency of the carbon budget approach. Coming back to the Green's function that is derived from the FAIR model and running it through the same optimization program, it provides more or less same path-dependent deviations, confirming that it can accurately capture path-independence at least in context of FAIR model. Therefore, we use the shape of Green's function to provide the intuition behind the path-independency and ground the argumentation that it can be used to assess other simple climate models' adherence to carbon budget (explained above).

Regarding the third point, I am happy to show the validation of GAMS in the revision in case permitted.

Finally, I would like to comment on ZEC. While I was aware of ZEC, I did not emphasize in my paper that TCRE is instantaneous, while ZEC is a measure for the temperature response after cessation of emissions. This is mostly visible by the not-so-well-managed term „net-zero" budget, which is equivalent to ZEC if we look at the temperature evolution in time further from the optimization year (Figure 2). The confusion and not precise usage of terms here come about due to the fact that this is an interdisciplinary paper, and to the best of my knowledge, ZEC is not yet established in the climate economics community. That does not, however, stop me from introducing it now and I would be very glad to have a chance to fix this in revision.

As a last comment, I would like to emphasize again that this work is an attempt to analyze the natural science of TCRE from a climate economics-optimized angle and also qualitatively aggregate and condense the findings for climate economic purposes, overall putting the work into a basket of interdisciplinary research.

References:

Dietz, S., van der Ploeg, F., Rezai, A., and Venmans, F.: Are Economists Getting Climate Dynamics Right and Does It Matter?, Journal of the Association of Environmental and Resource Economists, 8, 895–921, https://doi.org/10.1086/713977, publisher: The University of Chicago Press, 2021.

IPCC, 2013: Summary for Policymakers. In: Climate Change 2013: The Physical Science Basis. Contribution of Working Group I to the Fifth Assessment Report of the Intergovernmental Panel on Climate Change [Stocker, T.F., D. Qin, G.-K. Plattner, M. Tignor, S.K. Allen, J. Boschung, A. Nauels, Y. Xia, V. Bex and P.M. Midgley (eds.)]. Cambridge University Press, Cambridge, United Kingdom and New York, NY, USA.

IPCC, 2021: Summary for Policymakers. In: Climate Change 2021: The Physical Science Basis. Contribution of Working Group I to the Sixth Assessment Report of the Intergovernmental Panel on Climate Change [Masson-Delmotte, V., P. Zhai, A. Pirani, S.L. Connors, C. Péan, S. Berger, N. Caud, Y. Chen, L. Goldfarb, M.I. Gomis, M. Huang, K. Leitzell, E. Lonnoy, J.B.R. Matthews, T.K. Maycock, T. Waterfield, O. Yelekçi, R. Yu, and B. Zhou (eds.)]. Cambridge University Press, Cambridge, United Kingdom and New York, NY, USA, pp. 3–32

Meinshausen, M., Raper, S. C. B., and Wigley, T. M. L.: Emulating coupled atmosphere-ocean and carbon cycle models with a simpler model, MAGICC6 – Part 1: Model description and calibration, Atmos. Chem. Phys., 11, 1417–1456, https://doi.org/10.5194/acp-11-1417-2011, 2011.

Tokarska, K. B., N. P. Gillett, A. J.Weaver, V. K. Arora, and M. Eby, 2016: The climate response to five trillion tonnes of carbon. Nature Climate Change, doi:DOI: 10.1038/NCLIMATE3036.

---

## Author Comment (AC2)

First of all, I would like to thank the reviewer for carefully scrutinizing the manuscript, resulting in a very insightful and helpful review. Below I will reply to the reviewer's comments argument by argument, hereby highlighting the reviewer's elaborations in italic font. In particular, I will refer to the paper that was pointed out as missing literature. Indeed and regrettably, the Nichols et al. (2020) paper was overlooked, but it was a pleasure reading it and comparing it to my work. I believe that these two research pieces complement each other well, which provides motivation to improve my manuscript.

Before going into the reply, I would like to emphasize the key insights that my manuscript provides to the community. The manuscript provides the following novelties:

1. Suggests a simple method for future users to diagnose scenario-dependent deviations from the carbon budget approach for models of any complexity (not restricted to simple climate models).
   - From the pulse experiment, one can judge how scenario-independent TCRE is for fixed cumulative emissions
   - This has been shown in the manuscript in Sect. 3
   - The full model and Green's equation give the deviation in the same order of magnitude
2. It provides a formula for the state-dependent carbon budget approach, an alternative to the one provided by Nicholls et al. (2020).
   - In comparison to assuming the function in advance and then calibrating it to peak warming, the equation derived in the manuscript is motivated by thermodynamic properties —extract TCRE as a function of T, use a first-order Taylor expansion between TCRE and T, and then solve the differential equation.
   - In a sense, I provide an orthogonal approach that complements the one provided by Nicholls et al.
   - Both versions allow for both convex and concave relationship between cumulative emissions and temperature increase
3. I will show the qualitative behaviour of the pulse response under uncertainty in climate parameters (the demonstration is given in this reply)
   - By inspecting the behaviour of the pulse response under higher climate sensitivity, one can show that there is a limit to the carbon budget approach for a combination of high climate sensitivities and higher cumulative emissions.
   - The state-dependent equation under climate uncertainty is ready to be derived, allowing for expanding the utilization of Eq. 6 (manuscript) in the climate-economic assessments under uncertainty

Please find my replies as follows.

**"General comments"**

The reviewer captured the points of the paper correctly, with one correction on the sentence: „*They note that neither Green's formalism nor their new equation can capture scenario-dependence*". This is not quite the case. One can see that in the manuscript in Figures 1 and 2, right panels, that for various setups, Green's equation can indeed capture the scenario dependency of the full FaIR to a very high extent, especially in the mitigation scenarios (lower total $F_{tot}$).

Hence, one innovation the paper presents is that once a pulse response of a climate model was generated, Green's equation can predict the order of magnitude of the maximum possible scenario dependence for such a budget. Hereby, Green's method is not restricted to a "simple" climate model. One can, in theory, use the pulse response generated by ESM and feed it into Green's equation, run the min/max optimization (Eq 4), and get the deviation while avoiding running the full-fledged (!) ESM in the optimization scheme (which would likely be impossible due to computational costs). In essence, the manuscript suggests that Green's equation is a practical tool to check how the carbon budget approach works for any model. In a potential new version of the manuscript, this innovation would be made clearer.

**"Missing key literature":**

As I mentioned in the introduction, there was an oversight when scanning the literature resulting in claiming novelty in the state-dependent carbon budget equation (Manuscript, Eq 6). The reviewer points out that the equation serving the same purpose was derived in Nicholls et al. (2020), Eq. 4*. (I would like to reflect on Eq. 4* and my understanding of Nicholls et al. (2020) in order to compare Eq. 4* and Eq. 6, so I can simultaneously reflect on the questions raised in the review section). In the following, I reflect on the Nicholls et al. (2020) paper in high detail in order to clarify (what I see as) differences between the approaches.

Firstly, while reading Nicholls et al. (2020), it seems that the dots (Figs 1, 3 & 4) are referred to as both the peak warming and the warming at the time of the emission cessation. In my manuscript, I argue that those two are closely related. Additionally, looking at Fig 3, one can detect a large spread between the dots: the same cumulative emissions values give largely different temperature value (e.g. ~0.4 °C difference for ~1250 GtCO2). Unfortunately, I did not grasp whether the difference comes from different parameters, difference in non-CO2 forcers, or the warming definition (mentioned above). In any case, this leads to Eq. 4* and Eq. 6 not being comparable to a full extent, as Eq. 6 (manuscript) relates to the transient warming (not a peak warming) and also excludes non-CO2 forcers (as the reviewer already pointed out and I refer to it later in the reply).

Comparison between Eq. 6 (Manuscript) and Eq. 4* (Nicholls et al. (2020)):

Coming to the derivation of Eq. 4*, the authors assumed the logarithmic relationship between cumulative emissions and temperature as a starting point. Here, I question whether the logarithmic relationship between atmospheric carbon and radiative forcing is a valid ground to assume the logarithmic relationship between (cumulative) emissions and temperature; my suspicion comes from a convex relationship between cumulative emissions and atmospheric concentration that is a necessary link between emissions and temperature. Hence, the assumption is that a concave (logarithmic) relationship outweighs a convex one.
Coming back to my manuscript. Contrary to Nicholls et al. (2020), I do not assume a relationship in advance but derive it from approximating the pulse response with a TCRE that is dependent on the cumulative emissions and temperature increase.. Moreover, the method that I suggest (Eq. 5), does not require a pulse response at all, but one can map TCRE directly from the run (as temperature over cumulative emissions) and then proceed with deriving the equation. After mapping the TCRE that varies with temperature, I derive the equation using a physical quantity (temperature). Additionally, Eq. 6 can also be convex if parameter a changes sign. This has not been empirically observed in my experiment, but Eq. 6 does not theoretically restrict convexity. Furthermore, Eq 6. in the manuscript is strictly concave because only a single parameter combination (connected to the temperature saturation and carbon cycle weakening terms) was used. As I will show later, the pulse representation provides a hint about where (or when) the concavity in question breaks down. The discussion will show how the (manuscript) work continues with different parameter combinations, hence then including uncertainty.

To complete the comparison, I generated graphs inspired by Fig. 4b/d (Nicholls et al.) using the method and the runs from my manuscript. The following figures' left panels are to be compared with Fig. 4b, while the right panels are comparable with 4d. Note that the graphs are for demonstration purposes only.

[Figure]

At first glance, the right panels suggest that FaIR (manuscript) deviates much from linearity much more than MAGICC. This is not the case because I used GtC as an emissions metric and count from the preindustrial era, while Nichols et al. use GtCO2 and count from 2010. Therefore, the last figure (right panel RCP85) covers by far the highest range of cumulative emissions.

Furthermore, comparing the right panels with the Fig 4b in Nicholls et al. shows that Eq. 6 arguably emulates FaIR better than Eq. 4* MAGICC. However, this is no actual proof that my approach is more precise due to the much smaller emission pathway sample size.

Concluding the comparison (“Missing key literature“) section with the reviewer remark:„*Missing this paper (unfortunately) undermines the author's claims of complete novelty*.“. Regardless of a comprehensive scrutinization of the two approaches, the reviewer is, without a doubt, right about my statement of novelty. I would gladly revise the statements and incorporate new findings from Nicholls et al. 2020 in the revision.

**„Writing“:**

“*The paper reads like it hasn't been proof-read for basic errors (typos, missing words etc.) or overall structure and fluency*”. I truly regret that errors on this level occurred in the discussion paper. Measures have been taken that in case a revised manuscript is allowed for, that version will see a thorough proofreading by a native speaker. Thank you for pointing this out.

On the Heaviside function and its purpose in the text:
I fully see the reviewer's point about the triviality of the finding that the Heaviside function by itself gives the perfect budget, as it is (extremely) intuitive. The idea to put it in the text was not so much to show that perfect budget can be represented by the Heaviside function, but rather that using Heaviside in a Green's formulism is an intermediate, first-hand proof that Green's formalism is one way to describe the carbon budget equation in a general way (in which, I later show, it is possible to show scenario-dependent effects as mentioned in the “General comments” reply). Finally, indeed because of the pulse being close to Heaviside function (later after the relaxation), we have a very weak (and fainting) scenario-dependency of the cabon budget (Manuscript,Figs 1,2,3).
However, I see the logic behind putting this part in the appendix or the supplement and am happy to do so in case it is overall better.

“*In general, I would reconsider whether the more trivial observations would instead be better in a supplementary or simply removed (for example, the discussion of different ways one can reduce emissions and still have the same cumulative emissions in section 3.1.1)*.”
I am not sure if we mean the same thing, but the idea behind the end of that section was to show why for different run setups, we have different feasibility limits on k, which then effectively also affects the results, the closer we are to the feasibility limit.

The rest of the section on writing consists of very well points that should be incorporated in the revision in any case.

**“Key insights get lost“:**

Thank you for this valuable comment. I will expand Section 4 and probably focus much more on it in the new revision with fresh results that include uncertainty in parametrization (revealed in the “Lack of exploration of climate uncertainty” section of the review.

**“Journal fit“:**

Again, a very valuable comment. I was thinking about whether to keep the focus on the economics part and possibly change the journal, but I changed my perspective after introducing the uncertainty. I think discussing pulse representation with climate uncertainty fits the ESD and its audience.

**"Clarity of methods presentation":**

"*I would suggest removing discussion of the optimisation in the abstract*".
The reason for putting this in the abstract is to emphasize that this is a fully novel approach to calculating the carbon budget deviations when compared to previous literature. I found it important to emphasize that in this work, we do not test a portfolio of emissions but generate a maximally possible deviation – and hence the optimization. Maybe I could simplify it, but I still consider that it has its place in the abstract.

"*On this point, the open budget isn't clearly presented.*"
The condition $E(t>t*=0)$ is met only for the net-zero budget (I think this version goes in the same line as the budget considered in Nicholls et al) and not for the open budget (which would then go in line with the transient response interpretation). "*Which would seem odd, as then the cumulative emissions that the system sees are now changing from the specified limit up to t*.*" They are in fact, not changing, as they are set to be a fixed value ($F_{tot}$) at the time t*. The only thing that this condition does is that it allows (or constrains) emitting after t*, which is reflected only in the fact that because of the slope restriction, the net-zero budget requires the emissions to decrease prior to t* to reach zero at t*, while the open budget does not since then they can take any value at t*.

**„Lack of exploration of climate uncertainty":**

In this section, I can finally show what I was referring to in the sections above; I will show preliminary graphs that depict how I expanded this work to include the uncertainty in the climate response.

To represent uncertainty, we use 20 different parametrizations (lp1, lp2, … lp20) that cover 20 different equiprobable climate sensitivities in accordance with the IPCC AR6 range fitted to log-normal distribution (Forster et al., 2021). Lp1 represents the lowest climate sensitivity (CS), and lp20 the highest.

In the original manuscript, Figure 4 showed the pulse response of one parameter setup under different climatic conditions (different years in the RCP6 run). Conversely, the next figures show the pulse responses of different parameter setups (lp's) in one year of the RCP6 run.

[Figure]

A few effects can be detected. Firstly, one assumes that using the same method as provided in the paper, one can derive Eq. 6 dependent on CS too, with mapping TCRE (lambda) with both corresponding temperature and CS (not only temperature as in the manuscript).

However, the current hypothesis is that this method will not work for higher CS values (especially not lp20 for example). To understand it, remember that the state-dependent TCRE is an approximation of a pulse response in time when the pulse already reaches relaxation (near constant phase). As we can see in the figures above, the combination of higher climate sensitivities and different climate conditions (later years in the RCP6 run), the pulses change the behavior: instead of reaching a relaxation, they shift the trend towards increasing response in time. Hence, the method for deriving Eq. 6 is not compatible anymore.

This effect connects to the discussion about the convexity of the equation in the first review section. The reviewer states that the concavity in the carbon budget equation diminishes towards linearity (or even convexity), depending on the parametrization. In the pulse response representation (figure above), this can actually be detected in the mentioned shifting response trend. In essence, both higher climate sensitivity values (higher lps) and "higher" climatic conditions (later years of the run) mutually influence the system to be more linear (flattening the pulse response) until some threshold is reached, after which the response starts increasing later in time (most visible for lp20 where this effect is present in every year).

Following up on these findings and keeping in mind that the relaxation of the pulse is necessary for the carbon budget to be meaningful, we add one more finding. The new analysis with uncertainty provides insights into the values of climate sensitivity for which FaIR allows for a carbon budget approach at all. The following figure maps all of the pulse runs for all of the years and all of the CS values, calculating the proximity to the carbon budget (pulse relaxation) for every combination.

[Figure]

We can see how the carbon budget adherence changes with year of the pulse and climatic condition. This is still a work in progress, but in my view, it is moving towards a final product for the ESD.

**„Validation of model and code":**

I am happy to upload the code.

**„Claims about FaIR being 'best' ":**

I must admit that this was just a poor choice of wording. The authors suggest using FaIR carbon cycle in combination with the 2-box temperature cycle (effectively FaIR), but never state that it is the best. I think I would follow the reviewer's advice and simply remove the assessment since, as reviewer suggests, it is not relevant for the paper.

Concerning introducing negative emissions, I thank the reviewer for the hint about FaIR's ability to capture them. Depending on the scope of the paper, I will see whether and how to include them in the analysis.

**„Lower cumulative emissions level":**

Introducing the lower cumulative emissions level could be done but the analysis would be limited due to lesser feasibility (the feasibility limit is discussed in the contested redundant text at the end of 3.1.1.). Changing the way the emission rate is restricted (non-linear restriction) could maybe help in this case.

**„Leftover temperature":**

Regarding plotting, I think it is a good idea -- it makes perfect sense to include T_left in the supplementary.
"*Could you just run the Green's model over the historical period too*?" Not really, since I generate a pulse response used in Green's equation in present day conditions. The pulse response function also changes if we go back in history (like ti changes when we project it into the futue, Fig 4, albeit other way around). This means that we would also get some absolute temperature difference between full model and Green's equation.

**„Claims about time evolution of T_d":**

Regarding "*However, I am not convinced that this is a general thing rather than just being a model feature of FaIR, which appears to always reaches the same equilibrium for the same*". In a sense this was exactly my point when extrapolating the conclusion to other simple climate models. The Green's representation (using a pulse response function) shows whether a climate model can or cannot adhere to the carbon budget approach (and declining of the T_d). Hence, I claim that other models that do not reach relaxation in the pulse response will not adhere to the carbon budget.

Considering the second paragraph, I think I already reflected on most of the points of the reviewer by now. Regarding the non-CO2 forcers, I purposely left them out because I constrained my research only to carbon budget effects which, to my understanding, are not applicable to other forcers.

**„Why can't you just use a simple climate model":**

Firstly, the TCRE concept is of interest in its own right. In addition, within climate economics, the implementation of a model of FaIR's complexity represents a hurdle not every research group is willing to invest in. This holds more so for MAGICC. Finally, the TCRE concept represents an analytic bridge between two target-based decision-analytic concepts: cost effectiveness analysis (CEA) and cost risk analysis (CRA). CEA is a dominant paradigm in IPCC ARs 5-6, WGIII. However, it conceptually cannot deal with decision-making under anticipated future learning, as already pointed out by Blau (1974). For that reason, it might be accused of tackling an ill-posed problem for what reason Schmidt et al. (2011) modified it into CRA as one option for a 'repaired' version of CEA. Held (2019) raised the question of under what circumstances CEA-based scenarios can be retroactively justified as good approximations of CRA-based solutions, hence are well-posed under uncertainty. In Held (2019), a set of sufficient conditions is presented, which includes that temperature can be predicted by cumulative emissions (not necessarily linearly). Hence, for a justification of CEA in integrated assessment, the level of accuracy of TCRE for prediction is of high relevance.

"Figures":

As requested, here are the plots that correspond to the three runs in Fig. 6, depicting a **relative** deviation from full FaIR (% of the absolute temperature) for each model choice. Thank you for this suggestion, it is indeed quite helpful to get a clearer idea of what is going on.

[Figure]

References

Blau RA (1974) Stochastic programming and decision analysis: an apparent dilemma. Manage Sci 21(3):271–276

Forster, P., Storelvmo, T., Armour, K., Collins, W., Dufresne, J. L., Frame, D., ... & Zhang, H. (2021). The Earth's energy budget, climate feedbacks, and climate sensitivity.

Held, H. (2019). Cost risk analysis: Dynamically consistent decision-making under climate targets. *Environmental and Resource Economics*, *72*(1), 247-261.

Nicholls, Z. R. J., Gieseke, R., Lewis, J., Nauels, A., & Meinshausen, M. (2020). Implications of non-linearities between cumulative CO2 emissions and CO2-induced warming for assessing the remaining carbon budget. *Environmental Research Letters*, *15*(7), 074017.

Schmidt, M. G., Lorenz, A., Held, H., & Kriegler, E. (2011). Climate targets under uncertainty: challenges and remedies: A letter. *Climatic change*, *104*, 783-791.

---

## Author Response (AR1)

Dear Prof. Kirk-Davidoff,

hope this letter finds you well. I am happy to submit the revised manuscript following the reviewers' suggestions.

To my understanding, I have reflected on all raised issues and incorporated them into the manuscript. As suggested, I have shifted the article's focus more toward the climate modeling subject, expanding on the pulse response examination given in Section 4 and adding the effects of climate uncertainty. Nevertheless, I kept some references to the climate economics audience since I believe the work is of value in that domain.

Three main findings are reiterated here:

1.) Developed a scheme to predict maximum carbon budget scenario-dependency from the impulse response function, applicable to the models of any complexity

2.) Quantified maximum possible carbon budget scenario-dependent deviations

3.) Connected the behaviour of pulse response with the carbon budget non-linearities, while deriving an alternative formula for the carbon budget equation based on T as a predictor.

Once again, thank you for giving me an opportunity to revise the article. I hope that the reviewers will find the revision worthy, and looking forward to their fresh input. Please find the point-by-point response to reviewers, in parts motivated by the answers in the discussion phase. The file ends with a list of changes. I skipped listing all the changes in text because that would be an overflow of text, as one can see by the amount of the changes tracked in the "track-changes" file.

Best regards,
Vito Avakumović

Point-by-point response:

Reviewer 1

General Comments

**1)** The first general comment argues that the deviation from the linearity is only observed in simple climate models and EMICs, while at the same time, the CMIP5 models in Tokarska et al. 2016 (Figure 3) show the linearity. To this, I have two arguments. In Tokarska et al. 2016, the CMIP5 models were forced with both 1% per year atmospheric concentration increase forcing (pure carbon dioxide emission forcing) and the extended RCP8.5 scenario (aerosols and other non-$CO_2$ gases). The review argues that Tokarska et al. 2016 clearly show the linearity between cumulative emissions and the temperature increase with solid lines. However, those are the results using RCP8.5 scenarios so they have other non-$CO_2$ effects included, whereas FAIR was forced only with $CO_2$ in my work. Additionally, it is questionable if we have a linear relationship in a solid line for BCC-CSM 1.1 (green) in the relevant domain (~2000 GtC). On the other hand, it is not clear from the figure if the relationship is linear (or non-concave) for the purely $CO_2$ experiment (dotted lines). To round up, I quote from the paper „*Figure 3a shows that the warming in the RCP 8.5-Ext simulations scaled by the ratio of CO2 to total forcing, for a given magnitude of cumulative emissions, is slightly higher than for the 1PCTCO2 simulations. One possible reason for this is the warming from non-CO2 greenhouse gases, which reduces the diagnosed cumulative emissions in the RCP 8.5-Ext*

*simulations associated with the carbon-climate feedback.*" A similar conclusion can be made by looking at the first figure that I attach below, which I refer to in my second argument.

With that in mind, I come to my second argument. Looking at the graph given in the summary for policymakers AR5 WG1 (IPCC, 2013) in the first figure below, one can detect a slightly concave relationship between cumulative emissions and temperature increase, for RCP scenarios and the stylized 1% per year concentration increase scenario, also conducted in CMIP5 models. The figure can be found below. I drew the green and white lines to visually emphasize the deviation from linearity. In my research, only $CO_2$ emissions are taken into account, so the grey plume in Figure SPM.10 is imperative in this case. I use the result from the old report since in the newest SPM WG1 (IPCC, 2021), the relationship between cumulative emissions and the temperature increase is shown only for SSP scenarios (Figure SPM 10.). Additionally, the exponential relationship between cumulative emissions and temperature derived in the paper only slightly deviates from linearity in the cumulative emissions regime that is inspected. This was not shown in the paper not to overflow the reader with plots and information, but it can be tested with the values provided in the text.

Albeit the relationship is nearly linear, the small and persistent difference in temperature output can make a large difference in economic assessments.

Moreover, if the FAIR model does produce a higher deviation from linearity than the ESMs, one of the points of deriving the new equation (Manuscript, Eq. 6) was to show that one can fairly well approximate a FAIR climate model with a single equation if the state-dependent TCRE is considered (derived from FAIR). This is a result independent of the linearity discussion.

Lastly, in the reivsed manuscript I have shown a way how the calibration of FaIR model effects the linearity of the carbon budget approach through the pulse response representation. Indeed, it is possible to calibrate FaIR to have a linear relationship (and also convex) between cumulative emissions and temperature increase.

[Figure]

[Figure]

[Figure]

To conclude the point, the second reviewer pointed out the piece of literature that shows the non-linear carbon budget equation (Nicholls et al. 2020). Hence, previously published literature argues in favour of a possibility of non-linear relationship between cumulative emissions and temperature increase.

**2.)** On the second point, it was never argued that Green's function is an approximation of the real climate system, but rather an approximation of a model; in this case a FAIR model. Furthermore, the point of the paper was not to qualitatively detect the source of deviation from the real-world (natural) system, but to quantify what are the maximally possible deviations given the FAIR model, at the same

time testing whether FAIR (and its Green's approximation) can accurately capture path-independence. Hereby, I refer to the comments questioning both models' ability to capture the path independence. In my paper, the path dependence was tested by the optimization program given by Eq. 4. By using the optimization procedure, the full portfolio of all the possible emission pathways (under the given constraints) was tested, and not just „a simple series of idealized experiments using different rates of emissions", as suggested in the review. Hence, using the optimisation program the pathways that generate the minimal and maximal temperature in a given year ($y^*$) under the same cumulative emissions are generated. By subtracting those two, the maximal path-dependent deviation of the carbon budget is generated. Seeing it is small in magnitude, the conclusion is that the FAIR adheres to scenario independency of the carbon budget approach. Coming back to the Green's function that is derived from the FAIR model and running it through the same optimization program, it provides more or less same path-dependent deviations, confirming that it can accurately capture path-independence at least in context of FAIR model. Therefore, we use the shape of Green's function to provide the intuition behind the path-independency and ground the argumentation that it can be used to assess other simple climate models' adherence to carbon budget (explained above).

Additionally, the introduction of the Green's function provides a method to assess the carbon budget scenario-dependency of model of any complexity, not just simple climate models, which is one of the main novelties in the manuscript and offers a venue for further research.

**3.)** This point is addressed by providing all of the model runs and codes in open access online. My main concern in this validation request was that it is not specified which runs should be used to validate the GAMS translation. The logical choice would be the pulse response run. However, the inspection of the pulse responses was exactly the reason why I used GAMS and I do not see a way to do it in FaIR. Clarification is the following. To produce a pulse response one needs to set atmospheric carbon dioxide concentration to a fixed value and then generate the emissions that keep that level constant. GAMS makes this easy because in it, one only needs to change the role of emissions into a free variable (from being an input variable), and change the role of atmospheric concentration from a free variable to an input variable, and run the code normally afterward. In python this is not possible since one needs to have an explicit expression of how concentration depends on emissions in a functional form. FaIR, however, is constructed such that the concentration is implicitly dependent on emissions.

**4.)** This point was addressed in the revised manuscript in lines (260-265). Transient path-dependence is not independent of ZEC because the effect of ZEC cancels out when subtracting the max. and min. generated temperatures. Regardless, the implications of the work on ZEC are being addressed in the discussion in the revised manuscript (lines 620-628).

**Specific comments**

All of the specific comments given by referee 1 are now reflected upon in the revised text.

**Reviewer 2**

**General comments**
The reviewer captured the points of the paper correctly, with one correction on the sentence: „*They note that neither Green's formalism nor their new equation can capture scenario-dependence*". This is not quite the case. One can see that in the manuscript in Figures 1 and 2, right panels, that for various setups, Green's equation can indeed capture the scenario dependency of the full FaIR to a very high

extent, especially in the mitigation scenarios (lower total $F_{tot}$). Hence, one innovation the paper presents is that once a pulse response of a climate model was generated, Green's equation can predict the order of magnitude of the maximum possible scenario dependence for such a budget. Hereby, Green's method is not restricted to a "simple" climate model. One can, in theory, use the pulse response generated by ESM and feed it into Green's equation, run the min/max optimization (Eq 4), and get the deviation while avoiding running the full-fledged (!) ESM in the optimization scheme (which would likely be impossible due to computational costs). In essence, the manuscript suggests that Green's equation is a practical tool to test the carbon budget scenario-dependency for any model.

**Major concerns**
**Missing key literature**
There was an oversight from my side when scanning the literature, resulting in claiming the novelty in the state-dependent carbon budget equation. This mistake has been undone, and I have contextualized my work with the work from Nicholls et al. (2020). Moreover, referee's 2 comment had helped me to expand my work and gain new insight, which was incorporated in the revised manuscript. In my answer to the reviewer in the discussion phase, I have extensively compared my work in context of Nicholls et al. (2020) work. In essence, the method provided in the manuscript gives an alternative way to derive the non-linear carbon budget equation. Starting from the pulse response function, we approximate it using T as a leading variable (of a thermodynamic system) using a first order Taylor expansion, and then with integration arrive at the equation. In newly introduced Sect 4.4., I show how different calibrations of FaIR bring about different pulse response behaviour. Following the method given in the original manuscript, one can derive carbon budget equations with different levels of non-linearities (not restricted to convex relationship only) and even the linear equation, depending on the model calibration. This fits to claims from previous literature but seen from the fresh perspective (pulse representation)

**Writing**
The newly revised manuscript has been given proof-read by the native english speaker, so hopefully the text reads better. Also, I followed the referee's advice and removed much of the text that could have been seen as redundant or "slow", and made the language more active.

**Key insights get lost**
Following the referee's advice, I have revised the whole manuscript in order to expand on the analysis in Section 4.

**Journal fit**
The revised manuscript now focuses more one climate modelling tools, while still referring to the implications that the work has for the climate economic audience.

**Minor concerns**
**Clarity of methods presentation**
The revised manuscript still has a reference to the optimization in the abstract, but I have put it in a context. I think it is important to keep it, since it is the key point that differs how the scenario-dependent effects are examined in the manuscript, compared to the previous literature.
The open-budget and net-zero budget are renamed to fit more in the literature lingua. As pointed above, I have now made clear distinction between scenario-dependent effects explored in the manuscript, and ZEC.

**Lack of exploration of climate uncertainty**
The revised manuscript now introduces the effects of the climate uncertainty on the pulse response representation, and consequently on carbon budget deviations.

**Validation of model and code**
I have referred to this in the reply to the first reviewer (above).

**Claims about FaIR being 'best'**
I have taken the reviewers advice and modified the statements in Sect 2.1.

**Full-fledged**
The full-fledged noun has been removed and replaced with full model, so there is a distinction between the Green's and the model itself.

**Lower cumulative emissions**
I have introduced lower cumulative emissions that would be in line with 1.5 °C in the subsection that briefly investigates the effects of including negative emissions into the portfolio.

**Leftover temperature**
$T_{left}$ is prescribed only to show that Green's model can replicate the overall temperature increase. In context of testing scenario-dependent effects, it is not needed.
Moreover, running the Green's model over the historical period would not yield into correct temperature diagnosis, because the pulse response changes with changing climatic conditions (its magnitude decreases with higher background temperatures as shown in the revised manuscript Fig. 1). Hence, using the current-day pulse response as Green's function backwards in time would yield in underestimation of temperature, the same way it yields in overestimation later in the run with higher background temperatures (Fig. 5, top row).

**Claims about time evolution of T_d**
$T_d$ declining to zero is indeed the feature of FaIR, which is itself a good emulator of more complex models. This is why I expanded the analysis with inclusion of the one-box model which has a drastically different pulse response behaviour. Inspecting those two separately solidifies the conclusions from Sect 4. that claim that the pulse response can be used as an indicator of carbon budget deviations.

**Claims about impact of optimisation year**
I have backed up this claim with the examples given in the newly introduced supplement.

**Why can't you just use the simple climate model**
Technically, you can. My original point was to show that one can simplify the model even further, making it even easier to implement. More importantly, simple climate models in their present form are not fit for analytical inspection because solving them explicitly takes a lot of effort and provides cumbersome expressions. Hence, the idea was to give the audience a simple, analytically tractable expression that connects temperature and emissions.
Nevertheless, the focus of the revised manuscript is completely shifted so this discussion is not a vital part of it anymore.

**Figures**
I have taken referee's advice and plotted the relative deviation of the linear and newly introduced non-linear carbon budget equation from FaIR temperature diagnosis. It can be seen in the revised manuscript in Figure 8, bottom row.

**List of changes**
- Overall, profoundly changed the text in all sections with drastic text cuts and newly added text
- Emphasized the key points that seemed to be obscured to some extent in the original manuscript
- Expanded Sect 4., making its findings a new central point of the manuscript
- Pulse response representation as a focal point of the revised manuscript
- Included another simple climate model in the analysis
  - Confirms the findings in the light of the pulse response
  - Tests the deviations in the context of structural model uncertainty
  - Solidifies the pulse response (in the role of Green's function) as the viable carbon budget deviations indicator
- Included the effects of negative emissions on the carbon budget scenario-dependent deviations
  - On top
- Included the emission pathways that generated min. and max. temperature in Sect 3, and the resulting temperature pathways (not restricted to the optimization year only)
- Investigated the implications of climate uncertainty
  - Argued how different calibrations of FaIR produce different pulse response representation, from which both non-linearities and scenario-dependency effects of the carbon budget equations can be inspected
- Incorporated the findings from Nicholls et al. (2020), and contextualized it to the findings of the manuscript
- Put ZEC in the context of the manuscript and the calculated deviations
- Connected ZEC with the pulse response, offering a venue for further research
- Added the relative deviation of the newly introduced non-linear equation generated temperatures (and linear equation generated ones), from FaIR-generated temperatures
- Included the plots of temperature leftover terms corresponding to emission cessation in four different years
- Generally fixed and changed the nomenclature
  - Changed the lambda in Eq. 1 to uppercase, and renamed it from "perfect budget", to simply "linear" carbon equation
  - FaIR
  - Transient budget instead of open budget, removed S1 and S2 as defining the two cases
  - Historical scenario -> idealized (RCP6.0) scenario
- Fixed the ordering of the figures so they appear by order of appearing in the text
- Generally, fixed the visuals of the figures
- Changed the discussion
  - Removed the repetitive parts and offered a few venues for further research
- Fixed the typos

**References**

Dietz, S., van der Ploeg, F., Rezai, A., and Venmans, F.: Are Economists Getting Climate Dynamics Right and Does It Matter?, Journal of the Association of Environmental and Resource Economists, 8, 895–921, https://doi.org/10.1086/713977, publisher: The University of Chicago Press, 2021.

IPCC, 2013: Summary for Policymakers. In: Climate Change 2013: The Physical Science Basis. Contribution of Working Group I to the Fifth Assessment Report of the Intergovernmental Panel on Climate Change [Stocker, T.F., D. Qin, G.-K. Plattner, M. Tignor, S.K. Allen, J. Boschung, A. Nauels, Y. Xia, V. Bex and P.M. Midgley (eds.)]. Cambridge University Press, Cambridge, United Kingdom and New York, NY, USA.

IPCC, 2021: Summary for Policymakers. In: Climate Change 2021: The Physical Science Basis. Contribution of Working Group I to the Sixth Assessment Report of the Intergovernmental Panel on Climate Change [Masson-Delmotte, V., P. Zhai, A. Pirani, S.L. Connors, C. Péan, S. Berger, N. Caud, Y. Chen, L. Goldfarb, M.I. Gomis, M. Huang, K. Leitzell, E. Lonnoy, J.B.R. Matthews, T.K. Maycock, T. Waterfield, O. Yelekçi, R. Yu, and B. Zhou (eds.)]. Cambridge University Press, Cambridge, United Kingdom and New York, NY, USA, pp. 3–32

Meinshausen, M., Raper, S. C. B., and Wigley, T. M. L.: Emulating coupled atmosphere-ocean and carbon cycle models with a simpler model, MAGICC6 – Part 1: Model description and calibration, Atmos. Chem. Phys., 11, 1417–1456, https://doi.org/10.5194/acp-11-1417-2011, 2011.

Nicholls, Z. R. J., Gieseke, R., Lewis, J., Nauels, A., & Meinshausen, M. (2020). Implications of non-linearities between cumulative $CO_2$ emissions and $CO_2$-induced warming for assessing the remaining carbon budget. *Environmental Research Letters*, *15*(7), 074017.

Tokarska, K. B., N. P. Gillett, A. J.Weaver, V. K. Arora, and M. Eby, 2016: The climate response to five trillion tonnes of carbon. Nature Climate Change, doi:DOI: 10.1038/NCLIMATE3036.

---

## Referee Report (RR1)

**General comments**

Overall, I'd like to thank the author for their efforts revising the paper. The individual ideas are now much clearer and better explained. In particular, I think the ideas about what causes deviations from linearity and how these can be examined through impulse response are now much easier to grasp.

Some other especially nice parts:

- lines 49 to 53
- Figure 4
- line 402-413

I also think the paper opens up some nice follow up research on negative emissions and applying these ideas further.

Having said that, I still have some concerns, which I outline below.

**Major concerns**

**Key point of the paper**

When re-reading the paper, it wasn't clear to me what it's one, key point was. A few things seemed to potentially be the key point, the ones I could see:

- For economics applications, something easier to implement and understand than a simple climate model is needed. Here is a set of analysis that shows that a pulse response/Green's function based approach is a good approximation, some regimes to be aware of where the pulse response might start to break down (related to state- and scenario-dependence) and a non-linear equation for the relationship between temperature and cumulative CO2 that better captures state-dependence (which could also be useful where that state-dependence is important).
- Here we show how to understand non-linearities in the relationship between cumulative CO2 emisssions and temperature from the point of view of Green's functions/impulse response.
- Here we provide a method that allows us to predict a model's level of deviation from linearity between cumulative CO2 emissions and temperature based on knowledge of its pulse response alone

If any of these are the points, I think there are things missing. Respectively:

- The paper doesn't really go into enough detail to actually give climate economists enough information to understand how using a Green's function approach breaks down and under what conditions and how large this breakdown is compared to other uncertainties (for example, simply in just the size of the TCRE), particularly given how thin the section on climate uncertainty is (perhaps there is enough to tell a climate economist how much error is being introduced using the same parameter set the author used in this paper, but there is very little that could tell an economist how

big the error is if they assume a TCRE at the top of the IPCC range, for example).

- If showing how to understand non-linearities in the relationship between cumulative CO2 emisssions and temperature from the point of view of Green's functions/impulse response is the point, then build up the paper that way and use the optimisation and comparison to FaIR as validation, rather than starting from optimisation and working the other way (which is much more confusing in my opinion). If this is the case, I would cut most of the climate economics stuff and save that for a future paper (given what we know about deviations, this is what it means for climate economics).
- If you want to provide a method that allows us to predict a model's level of deviation from linearity between cumulative CO2 emissions and temperature based on knowledge of its pulse response alone, then you need to verify this across a much larger part of FaIR's parameter space and arguably with many more models (or you have to caveat your conclusions appropriately). Further, if this is the intended point, I think some analysis to investigate the number of impulse response experiments required to make robust conclusions would be important (given that the number of impulse response type runs we could do with ESMs will be limited).

I think this concern could be thought about another way. In short, after I read this paper, what should I come away thinking? Should I think that we absolutely have to included non-linearities when doing carbon budget calculations? Should I reconsider using a simple pulse response in my climate economics work? Overall, I read the paper and thought, "A lot of this is quite interesting", but I had no idea what I was meant to takeaway from it nor what the key message was.

Fixing this point would also greatly help the paper's communication. It is greatly improved from the first iteration, but the paper is still quite slow. If the paper's point were clearer, then much of the text could be cut (or moved to the supplementary, if the author can't bear to part with it) because it would be clear that it wasn't directly relevant to the main point.

**State-dependence of TCRE**

Assuming TCRE = -a T + b seems to not be a great choice to me. Specifically, its limits don't make sense. It seems an odd idea to me to suggest that TCRE could to zero (or even negative), particularly because if TCRE is zero then you can neither warm nor cool, no matter how much more CO2 you emit or remove. Can you re-think this functional form or put some limits on the domain of applicability you think it could have.

**Minor concerns**

**Overblown conclusions**

There are only a couple of these, but I would strongly encourage the author to be more cautious with their wording when making conclusions. A key example

is in the abstract (line 18-20),

"The analysis shows that using the Green's function approach to diagnose a model's carbon budget scenario-dependency, along with the method of deriving the non-linear carbon budget equation, both do not depend on the complexity of the chosen climate model."

There is absolutely no way you can conclude that using FaIR and the one-box model. They are far too simple. Computational constraints mean we can probably never make a statement like this, because we'll never be able to do the experiments with an ESM. Something like the following would be an appropriate softening,

"The analysis finds that using the Green's function approach to diagnose a model's carbon budget scenario-dependency, along with derivation of a non-linear carbon budget equation, doesn't depend on the complexity of the two simple models used here, leaving investigation with other and more complex models to future work."

**Exploration of scenario space**

The author says that their use of optimisation means they explore a greater amount of the scenario space than other papers. That's probably technically true, but I think it is a bit of a stretch to say that this is a really novel aspect. Nicholls et al. (2020) used all the SR1.5 CO2 pathways, which cover an already wide range of different rates of mitigation considered plausible.

The optimisation goes further than this in terms of pathway exploration, but whether this is a sensible extension or not is questionable I think, particularly looking at the triangle shape of the pathways in Figure 4c and 4d, which I think prove that the author's constraints don't actually prevent unrealistic pathways.

I think it should be noted that the author also applies (arguably arbitrary) contraints on emissions in their optimisation. I think this undermines lines 95-96, "Through the optimization scheme, the full portfolio of emission pathways is tested." Given that a full portfolio can't truly ever be claimed, I would suggest removing lines like these, rather prefering statements like, "a wide range" or "range that can be explored freely by the optimiser".

I found the discussion in 4.1 far more convincing than the optimisation in terms of explaining how you can go from impulse response to scenario deviation. Having read this section, you could cut the optimisation completely and just construct, by hand, pathways that lead to this maximial scenario dependence based on the insights presented in Section 4.1.

In the author's reply, they said, "I think it is important to keep [the optimisation], since it is the key point that differs how the scenario-dependent effects are examined in the manuscript, compared to the previous literature". As I've said above, I don't think this element is that novel or key in terms of making this

manuscript stand out (and as I've said further above, it wasn't obvious to me that this was the key difference/point of this paper).

**Freely evolving case**

I still don't understand the freely evolving case. After t* there is no constraint, so couldn't your program just dump out 1000 GtC in a single year? That would cause quite different responses no? It would be helpful if you could explain how emissions after t* are decided in the case that there are no constraints on this period (which is what I understand the freely evolving case to be)? Or is the point if this freely evolving case that you remove the constraint that $E(t*) = 0$ but leave all other constraints the same? If yes, I think it would help to clarify this in the manuscript.

**T_left**

As far as I can tell, T_left is only required for comparison to FaIR, and cancles out in the calculation of T_d (because it is the same in both T_max and T_min). If that is correct, why is T_left introduced in 3.1.4 and not in the section where you compare the pulse response based method and FaIR (where it would seem to belong better as essentially a correction factor to deal with different start dates rather than something fundamental to your overall methods)?

**Dependence on optimisation year**

There is now a supplementary figure showing the deviation as a function of the optimisation year. However, it is only done for a single F no? Don't you need to vary both F and t* to make any conclusion of robustness in this relationship?

**Technical corrections**

- In response to reviewer 1's point 3, the author said, "However, the inspection of the pulse responses was exactly the reason why I used GAMS and I do not see a way to do it in FaIR." I will just note that it is possible to run FaIR concentration-driven in Python, you just have to do a bit of digging to find the configuration. If you raise an issue on their GitHub, I'm sure they will reply fairly quickly.

- line 61-63: "Regardless of ZEC, the linear segmented framework concept itself has been challenged by Nicholls et al. (2020), who claim that its assumption of a linear relationship between peak warming and cumulative emissions leads to unrealistically low budgets." I think you may have misunderstood the paper. When I look at it, the paper doesn't come to this conclusion at all, particularly given how wide the uncertainties are where they're presented in the main text. The paper has a look at the implications of including non-linearities and basically concludes, they're

pretty small in the context of other uncertainties so ignoring them in the segmented framework is not a terrible approximation.

- line 73-74: "Nicholls et al. (2020) have derived the non-linear carbon budget equation by positing a logarithmic relationship between cumulative emissions and temperature increase". This might be a slight mischaracterisation, it is a logarithm but it can actually go both ways so it isn't always logarithmic saturation (which is how this text reads).

- line 91-94: "At its core, this paper endeavors to define and assess both the scenario- and state-dependent deviations (non-linearities) of the carbon budget approach. It demonstrates that a temperature response to an emission pulse, i.e., the pulse response representation, offers a very convenient tool for doing so." I would blend this sentence into one because the deviations aren't novel to this paper, but the pulse response representation is (noting also the major concern about the key point of this paper being unclear as it is currently written).

- lines 102-104: "allows us to calculate the maximum possible scenario dependency of the ESM models" should be "allows us to approximate the maximum possible scenario dependency of the ESM models" (as the author says, you can never do this with an ESM because it costs too much so you're left with approximations at best)

- lines 113-114: "For climate economists, it reveals the consequences of using models with incorrect pulse representation, in terms of their inability to adhere to the carbon budget approach". If this is the point of the paper, make it more obvious (see also major concern).

- lines 148-150: "Specifically, the sets used in this paper are tuned to the MIROC-ES2L, BCC-CSM2-MR, MPI-ESM1-5, CNRM-ESM2-1, and ACCESS- 150 ESM1-5 models." You should cite the model description papers here (those model developers put huge effort into making the models available and citations like this are key for recognising that effort).

- line 182: "The difference is that, in Eq. (6)" Should this refer to Eq. (2)?

- line 185: "To make use of Eq. (2), one must opt for a shape matching Green's function fg." What does this mean (I think the words 'opt' and 'a shape matching' are the things that make this most unclear)? "One must find an appropriate Green's function"?

- Figure 1: where is "TCREv2" on this figure

- lines 192-193: "In the year of pulse response generation tp, the emission pathway necessary to keep the level of atmospheric concentration Ca(tp) constant is generated" Why do you need this? Keeping Ca constant leads to ongoing warming. Don't you want emissions that keep temperatures constant (or just add the pulse to the RCP emissions and don't worry about the fact that temperature is changing)?

- line 214: "would show deviations" -> "would not show deviations"?

- line 279: "again, independent of ZEC, as explained above". You get to this point later in the manuscript, where as you say it's only independent of ZEC if the ZEC is also path-independent, which I don't think it is so 'approximately independent of ZEC' throughout is probably better/necessary here (possibly also with a forward-reference to your later discussion)

- line 352: "a deviation of 0.15C is produced." What is this in percentage terms?

- line 357: "In the supplement material, various combinations of the same cumulative emissions and different t*'s show that the deviation not being a function of the optimization year is a robust result." Tell the reader which supplementary figure. Looking at the figure, it is only done for a single F? Don't you need to vary both F and t* to make this conclusion of robustness?

- line 393: "Fig. 1, left graph shows the FaIR-generated Green's function (blue)." You can delete sentences like this, they add nothing on top of what comes after. The context can simply go at the end of the point you're making. In this case, at the end of the next sentence.

- Figure 9d: something wrong? Red line in ACCESS disappears

- line 558: "Additionally, it would be interesting to see to which extent FaIR tuned to a CMIP6 model reproduces the behavior of its corresponding ESM under the same setup", Doesn't Leach do this as part of their analysis i.e. don't you already have the answer? Or do you mean something else?

- lines 617-619: "In the context of adhering to the temperature target, the declining temperature following emission cessation leads to non-intuitive policy recommendations, namely, to perpetually (albeit at a decreasing rate) continue emitting in order to adhere to the target." I'm not sure I agree with this? Doesn't this just mean that you've assumed the target is a stabilisation target whereas there is no precedent for that in e.g. the Paris Agreement (which isn't a stabilisation target, but rather a 'do not exceed' target, and arguably a temperature decline target given Article 4's wording). I would re-word this.

---

## Referee Report (RR2)

Nicely done. The paper makes its key point clear and offers a nice addition to the literature on the carbon budget. I congratulate the author for their efforts on the paper and on this last set of revisions in particular.

I have included a number of technical corrections below. There are only a few that are critical (i.e., I couldn't understand the sentences as currently written). The rest are mostly tidying. These can be included or not. It is really up to the author and editor how far they want to push the paper's quality/readability. I would tidy these things up because I found the writing issues super distracting, but ultimately it's up to you and the editor.

As said previously, well done on a great paper.

**Technical corrections**

- lines 3-5: I had a really hard time understanding the sentence. Is the following change an accurate reflection of what you meant? "In this paper, the deviations of the carbon budget and the strict linear relationship implied by the carbon budget are examined through the lens of the function of the temperature response to an emission pulse (i.e, pulse response), and its relation to TCRE" -> "In this paper, the deviations of the carbon budget from the strict linear relationship implied by the TCRE is examined through the lens of impulse response and its relationship with a non-linear TCRE."

- line 5: "deviations" -> "deviation"

- line 6: "The former stems from the scenario choice, the emission pathway, under the fixed cumulative emissions" -> "The former stems from the scenario choice, i.e. the specific emission pathway for a given level of cumulative emissions"

- lines 8-9: "This paper shows how the pulse response in role of a Green's function gives a unifying perspective on both scenario and state-dependence" -> "This paper shows how the pulse response, viewed as a Green's function, gives a unifying perspective on both scenario and state-dependence"

- line 10: "independency" -> "independence"

- line 10-11: "under the given constraints" -> "for a given set of constraints"

- line 12-13: The sentence as written is very confusing. Does the following change clarify the intended meaning? "Moreover, using pulse response as a Green's function in the optimization program, the" -> "Moreover, using the pulse response, the"

- line 13: "full" -> "reduced-complexity". ESM people would rightly have a fit if we called FaIR a full model.

- line 19-21: "Green's function approach is eligible to diagnose both models' carbon budget scenario-dependency, leaving investigation with other and

more complex models to future work." -> "The Green's function approach can be used to diagnose both models' carbon budget scenario-dependency, paving the way for future investigations and applications with other and more complex models."

- line 63: "with the changing" -> "with changing"

- lines 64-65: "Moreover, this paper shows that state-dependency of TCRE leads to a non-linear carbon budget equation, as the non-constant TCRE leads to breakdown of linearity given by Eq. 1." This sentence goes in circles (non-linear TCRE leads to non-linearity because it breaks the linear assumption). I suggest deleting it.

- line 67: "and not from the climate conditions of the system" -> "and not from the intial climate conditions of the system". If you have a different pathway, then you have different climate conditions along the way to reaching the same cumulative emissions so I don't think it is correct to imply that the climate conditions are entirely the same.

- line 68: "utilizing the high-complexity" -> "utilizing high-complexity"

- line 87: "of Green's function" -> "of a Green's function equation" ? The phrase "in the context of Green's function" doesn't make any sense to me.

- line 91: "independency" -> "independence" (again, probably worth doing a search for independency throughout, independency is an anarchic term for independence)

- line 92: 'full model' -> "FaIR" or whatever other model you mean. I wouldn't use the term 'full model' anywhere in your paper because it is unclear and the idea of a 'full model' doesn't really make sense (all models have limitations, so what makes one model full while another is not).

- line 99: as above re 'full model' (and, as above, I strongly suggest removing the phrase 'full model' from the manuscript entirely)

- line 108: "left out" -> "left"

- line 113: "in context" -> "in the context"

- line 114: "implications on" -> "implications for"

- line 117: as above re 'full model'

- line 123: as above re 'full model'

- line 125: "deviation" -> "the deviation"

- line 132: "changing pulse" -> "pulse" (changing is already implied by other words in the sentnce)

- line 167: "the Green's" -> "Green's"

- line 253: "45 years" -> "45 years farther" (or further)

- line 265: "previous" -> "the previous"

- lines 303-304: I didn't understand this sentence: "This way, assuming any function for the state dependency is avoided; rather, it is deducted from mapping v(T) (Fig. 7, right graph)"

- line 305: "due to negative" -> "due to the negative"

- lines 307-308: "due to an an increasing probability of a climate system bifurcation thresholds" What does this mean and what reference do you have for this?

- line 321: "shown in next" -> "shown in the next"

- line 359: 'full model' (as above)

- line 361: 'full model' (as above)

- line 366: refer to the analysis you have which shows this (Appendix A)

- line 415: "exhibiting the same" -> "exhibiting deviations of the same"

- line 416: "especially precise for lower cumulative emissions" -> "with the Green's function approach being especially close to FaIR for lower cumulative emissions"

- line 427: "Namely, in" -> "Namely,"

- lines 429 - 431: There are some missing words in this sentence so it doesn't really make sense

- line 490: 'had' -> 'has'

- line 495: 'full model'

- line 515: 'linearities' -> 'non-linearities' ?

- line 518: 'linear' -> 'linear (when viewed at the same time point)' or something like this? The pulse isn't constant so there must be some time element assumption in here no?

- line 540: "with FaIR" -> "with FaIR or some other model that captures the scenario- and state-dependent effects discussed here"

- line 662: 'full model' (as above)

---

## Author Response (AR2)

Dear Prof. Kirk-Davidoff,

Thank you for carefully reading the revised manuscript and for your helpful suggestions in view of further improving the manuscript. After thorough revision, I believe I have successfully addressed both reviewers' comments, as I will reflect on below.

The newly revised manuscript has mostly been built on the advice about clarifying the central point of the paper. As you suggested, the point is to demonstrate the usefulness of pulse response in the role of Green's function for diagnosing deviations from the carbon budget approach. The other two themes that Reviewer 2 candidates as possible central themes are much less salient, with the climate economics part (1st point) being only briefly mentioned, and the third point, which is generalizing the method of using pulse response for carbon budget deviations, brought in the paper as potential use, left for future work.

As per suggestion, Sect 3. and Sect 4. have now replaced the order of appearance with an optimization program in the role of validating the concepts introduced in Sect 3. However, I have kept the dual role of the optimization program, with the second role (besides Green's function validation) being the generation of maximally possible scenario-dependent deviations under given user-defined constraints. Although Reviewer 2 is not wrong that the scenario-independency stemming from plausible future emission scenarios has been tested, that is not the point of the optimization program. The optimization program tests the whole emission scenario space that could be considered possible. Even though it is not a large breakthrough compared to previous literature, it adds value to the literature by confirming scenario independence to a higher level. Additionally, as I suggested at the end of the discussion, the optimization program could be tested in other, more complex models for verification of previous results.

I have kept the one-box model pulse response representation because it explicates and confirms the pulse response view of carbon budget deviations, giving a counter-example to FaIR as a model whose pulse response suggests large and persistent deviations. Nevertheless, I have been more explicit this time in pointing out that the two models are not on par, and a lot of material in regard to one-box has been removed.

Finally, I have followed the reviewer's 2 advice in lines of cutting the text out / putting it into supplementary. The subsection "net-zero case" that was in Sect 3.2.1 (previous version) is now fully removed, while the section that introduces temperature leftover has been moved to the appendix. Supplementary material has been boosted with the optimization run detailed setup description.

I will refer to the rest in the point-by-point response, followed by this letter.

Once again, thank you for giving me an opportunity to revise the article again and for helpful guidance on how to do it the best way. I hope that the reviewers will find the revision in a good light, as I did my best to follow the given suggestions. The list of changes can be found after the point-by-point response.

Best regards,
Vito Avakumović

**Referee 1**

Overall Assessment:
I would like to thank the reviewer for the very kind and encouraging words in regard to the revision of the original manuscript. I hope the new version meets the same level of quality.

General comments:

- The font size is increased in all figures
- Units have been changed as suggested. Also, GtC is now presented as PgC.

Specific comments:

- The specific comments have been addressed in the text and the figures have been fixed
- Answer to the comment "*Figure 5: Would be clearer if used same y-axis range for all subplot in top row*".
  - While it is a valid point in general, I would disagree in this specific context. I chose a different y-axis range such that the central value (the magnitude of the absolute temperature increase) would always be approximately in the middle of the y-axis. Since the magnitudes are different for every graph, the y-axis changes. If I chose the same y-axis range for all the graphs, the difference in generated temperatures between FaIR and Green's approach for each case of F_tot would not be in the foreground, which is the point of these graphs.

**Referee 2**

**General comments:**

I would like to extend my gratitude to the second reviewer for a thorough examination of the second manuscript and for very detailed advice on possible ways to improve it. I believe the addressed comments in the second review phase made the manuscript a better version of what it was. The newly revised manuscript has been designed to address the major and minor concerns, and I sincerely hope that the latest version is clearer to read and more on point.

**Major concerns:**

**Key point of the paper**

The ambiguity about the key point has been hopefully resolved in the newest revision. The main point of the paper is indeed the second point, i.e., showing how to understand both state-dependent and scenario-dependent deviations of the relationship between cumulative CO2 emissions and

temperature (the carbon budget approach) from the perspective of the pulse response representation (in the role of Green's function). Hence, as suggested by the reviewer, the sections' order of appearance has been rearranged. In the revised version, the paper first shows how to understand deviations through the lens of pulse response in Sect. 3, followed by the validation of the theory and quantification of the deviations in Sect 4.

Furthermore, as suggested, most of the climate economics (point one) had been cut out of the text, with some brief referencing in the context of the pulse response representation and climate models that are used in climate economic assessments.

The last, third point was not intended to be a central point of the paper and is less emphasized in the revised version. The 'robust conclusions' are reduced, and generalizing Green's approach to models of higher complexity is given as a suggestion for future research, instead of as a novelty claim of the paper.

Lastly, the issue of "*the paper is still quite slow*" was addressed. A lot of the text has been cut out (for example, the whole section that was 3.2.1 in the previous version does not appear in the new version anymore)

**State-dependence of TCRE**

The functional form TCRE = - a T + b was not chosen but derived empirically from the interpolating the points of the state-dependent pulse response (approximated as state-dependent TCRE), as shown in Fig 2b (revised ms). I have addressed the reviewer's concern in the main text, where I put the limits of the equation backed by the emission runs under which the equation had been tested. Specifically, "*It seems an odd idea to me to suggest that TCRE could to zero (or even negative)…*", this is not the case with this equation, or at least not within any reasonable values of temperature. Checking the values of the coefficients a and b, one can see that TCRE reaches 0 with the temperature around 15 K, and goes to negative for larger values.

**Minor concerns:**

**Overblown conclusions**

"*A key example is in the abstract (line 18-20)*". The critique is valid, and it has been properly addressed in the revised manuscript. The discussion addresses potentially using the Green's approach with more complex models but with caveats that the validity of the approach cannot be tested in the ESMs since they will never be able to be run in the optimization program, because of their size. However, as an outlook at the end of the discussion section, I added the potential of using it as a more complex than FaIR, but still a climate model of reduced complexity in the optimization run and checking its pulse response under different parameterizations. Using the more complex climate model, but still a relatively simple one would be a first step towards verifying the findings of this paper further.

**Exploration of scenario space**

"*The author says that their use of optimisation means they explore a greater amount of the scenario space than other papers. That's probably technically true, but I think it is a bit of a stretch to say that this is a really novel aspect. Nicholls et al. (2020) used all the SR1.5 CO2 pathways, which cover an already wide range of different rates of mitigation considered plausible.*"

While I understand the reviewer's concern about overblown conclusion about the exploration of scenario space, I must point out that the reviewer himself states that it is technically true that the

optimization program gives means of exploring a greater amount of scenario space than other papers, meaning that it is to some extent novelty. The itself is that a bigger set of emission scenarios have been tested (in fact it is full emission space under given constraints), and moreover, the paper suggests a method for future research and different models to test the scenario-dependent deviations in a form of the optimization run, which could be useful for the community. Furthermore, I never argued in the paper that the diagnosed scenario-dependent deviations stem from plausible emission scenarios, but only that the optimization program tests the extreme possible cases under the given user-defined constraints. The fact that the optimization program tests possible, not necessarily plausible emission scenarios, is now explicitly stated in the revised manuscript.

Moreover,
*"I think it should be noted that the author also applies (arguably arbitrary) contraints on emissions in their optimisation. I think this undermines lines 95-96 "Through the optimization scheme, the full portfolio of emission pathways is tested.". "*
Again, the constraints are arguably arbitrary and attempted to be justified in the section that discusses the boundary conditions, which is now moved to supplementary material. To emphasize this point, I have emphasized throughout the text that the portfolio of emission pathways is constrained by user-defined constraints.

*"I think it is important to keep [the optimisation], since it is the key point that differs how the scenario-dependent effects are examined in the manuscript, compared to the previous literature". As I've said above, I don't think this element is that novel or key in terms of making this 3manuscript stand out (and as I've said further above, it wasn't obvious to me that this was the key difference/point of this paper)."*
I can only reiterate what I tried to convey in the points above (and in the letter to editor), and also the reply that I given in the first round of the discussions. However, the whole paper's focus is shifted away from the optimization program, and the optimization is now used foremost as a validation tool in a revised manuscript, so I hope that the compromise is found. I could technically remove it completely, but I still think it is a valuable result, as a tool to test maximally possible (not plausible) scenario-dependent carbon budget deviations under the given user-defined constraints.

**Freely Evolving case**

Thank you for pointing out this weak explanation; I was not fully aware of it myself. Indeed, the rest of the boundary conditions stay the same as only the boundary condition in the optimization year changes. This, however, drastically affects how the deviations behave since, because of the slope restrictions, the pathway in the net-zero approach must start declining ahead of time to reach 0, regardless of whether we minimize or maximize. This makes the minimization and maximization pathway more similar in net-zero, than the transient budget case.
Nevertheless, the comparison between net-zero and transient budget case have a much smaller role in the revised paper, as it focuses on the investigation of the pulse response instead. Hence, the distinction between net-zero and transient budget case is described in more detail in the supplement. In the main text, they are distinguished in one sentence with the one characteristic that differs between them, i.e., the condition of no emission in the optimization year for the net-zero case.

**T_left**

Once again, I can only commend and thank the reviewer for careful scrutiny of the paper. As the article is no longer centered around the optimization scheme, the whole section that was describing

the optimization scheme (including a misplaced T_left discussion) has now been removed from the main text. As such, T_left found its place in the appendix.

**Dependence on optimization year**

Both F and t* are varied in the revised manuscript.

**Technical corrections:**

- All of the suggestions in this section have been incorporated in the text. Some of the points are already covered in the discussion above.
- "*I will just note that it is possible to run FaIR concentration-driven in Python, you just have to do a bit of digging to find the configuration.*"
  - This is probably true since I never asked the authors directly for a FaIR concentration-driven
  - I guess the reviewer meant the version that can be found on the following link - https://github.com/OMSNetZero/FAIR/blob/master/src/fair/gas_cycle/inverse.py
  - However, the anecdotal reason why I haven't used it is the following:
    - The code generated at the link above was uploaded at the end of 2022, I started playing with the initial (in the meantime scratched) idea of this paper before at the beginning of 2022
    - Hence, I did it in GAMS, because it deals with changing the role of a parameter to a variable and vice versa rather easily as opposed to Python
- "*line 61-63…*" fixed! I emphasize in the text that the non-linearities are small in comparison to other uncertainties.
- "*line 73-74*…" By logarithm, I just meant a logarithmic functional form, which is the assumed functional form in Nicholls et al. (2020). I added a statement in the text that underlines that there is a multiplying factor that makes sure that the equation can take both concave and convex form (since stating that it is logarithmic can be indeed misleading).
- "*line 91-94*" blended.
- "*line 102-104*" discussed above
- "*lines 113-114*" This is not a main point of the paper, but it is of concern to a large part of the community that uses simple climate models in the climate economic field. Hence, I do not make it more obvious as a point of the paper (because the point of the paper is the pulse response discussed above), but I put it in a discussion section where I took the liberty to discuss the findings of the paper in a broader context. I admit that it is a subjective assessment to bring it up in the discussion section. I could easily remove this part, but I still consider it a very valid concern for climate economic discipline, worthy of mentioning.
- "*line 148-150*" Initially, I did not cite the authors since I used the parameters from the FaIRv2.0.0 paper (Leach et al. 2021), where I had trouble finding the citations. Nevertheless, I hope this is now fixed as I dug in and looked for the model description papers.
- "*Line 182*"/"*Line 185*" fixed
- TCREv2 is removed
- "*lines 192-193*" The procedure of acquiring the pulse response in the article follows the same procedure as the preceding literature that inspects pulse responses (albeit not to the same level of detail and not varying the climatic conditions as done in this paper), i.e., Joos et al. (2013) and Millar et al. (2017)
- "*line 279*" in the previous (second) version of manuscript, by this passage I meant that net-zero budget deviation is independent of ZEC because the ZEC effects subtract each other when subtracting T_max from T_min – the same way that scenario-dependent deviations (T_d's) are independent of T_left, as the reviewer already pointed out in his review
- "*line 352*" varied for both in the revised version

- "*Red line in ACCESS disappears*" fixed
- "*line 558*" Here, I meant that the pulse response experiments under different climatic conditions (referred to in the paper as the pulse representation, shown in Figs 1 and 3) were not (to my knowledge) done in the ESMs. So, it would be interesting to see to which extent the pulse response representation of FaIR corresponds to the ESMs pulse response to which it was calibrated.
  In other words, yes, FaIR was tested to reproduce CMIP6 behaviour in prescribed emission scenarios in Leach et al. (2021), but there is no explicit comparison between pulse responses under different climatic conditions.
- "*lines 617-619*" I removed these lines in the latest revision, as they referred to the lines from the introduction that talk about the carbon budget in the climate economic context (removed as well). A brief explanation of these lines: Namely, it refers to the decision-making framework used in climate economic calculations called "Cost-effectiveness analysis", where meeting a temperature target is ingrained in the optimization program in a sense that the climate-economic model finds the cheapest way to adhere to the climate target. As a consequence of mathematic formulation, the model will then try to stay as close to two degrees because sees it as a boundary condition. See Figure 5 in the supplementary material of (Neubersch et al. 2014)

**List of changes**

- All of the changes (except changing the y-axis in Fig 5, upper panels) suggested by the referees
- Changed the units from °C to K
- Adjusted the focus of the paper towards inspecting the pulse response dynamics and its implications on the deviations of the carbon budget approach
- Removed the carbon budget implications on and referring to climate economics from the introduction
- Changed the sections' order of appearance (Sect. 3 and 4. switched and modified)
- Moved the detailed description setup of the optimization run and the choice of boundary conditions in the supplement
- Removed the "net-zero" optimization scheme results section, Sect. 3.2.1.
  - Used net-zero results in the last section of the paper that discusses the transitory nature of the scenario-dependent deviations
- Moved the temperature leftover modification on the Green's approach section to the supplementary
- Added more figures (varying F_tot and t*) showing the optimization year independence in the supplementary
- Introduced the domain of applicability of TCRE(T) relationship
- Changed the Conclusion, so it does not refer to sections anymore, but to overall topic of the paper
- Rephrased the claim in the introduction about Nicholls et al. (2020) showing that linear equation leads to unrealistically low budgets and fixed the claim about their logarithmic relationship
- Fixed the "overblown conclusions" issue, while keeping the optimization scheme in a twofold role: primarily as a validation for the Green's approach, and secondary as a novel tool of inspecting the possible (not plausible) range of maximal possible scenario-dependent deviations under user-defined constraints
- Increased the font size on all of the figures

**References**

Joos, F., Roth, R., Fuglestvedt, J. S., Peters, G. P., Enting, I. G., Von Bloh, W., Brovkin, V., Burke, E. J., Eby, M., Edwards, N. R., et al.: Carbon dioxide and climate impulse response functions for the computation of greenhouse gas metrics: a multi-model analysis, Atmospheric Chemistry and Physics, 13, 2793–2825, 2013.

Leach, N. J., Jenkins, S., Nicholls, Z., Smith, C. J., Lynch, J., Cain, M., Walsh, T., Wu, B., Tsutsui, J., and Allen, M. R.: FaIRv2.0.0: a generalized impulse response model for climate uncertainty and future scenario exploration, Geoscientific Model Development, 14, 3007–3036, https://doi.org/10.5194/gmd-14-3007-2021, publisher: Copernicus GmbH, 2021.

Millar, R. J., Nicholls, Z. R., Friedlingstein, P., and Allen, M. R.: A modified impulse-response representation of the global near-surface air temperature and atmospheric concentration response to carbon dioxide emissions, Atmospheric Chemistry and Physics, 17, 7213–7228, https://doi.org/10.5194/acp-17-7213-2017, publisher: Copernicus GmbH, 2017.

Neubersch, D., Held, H., & Otto, A. (2014). Operationalizing climate targets under learning: An application of cost-risk analysis. Climatic change, 126, 305-318.

Nicholls, Z., Gieseke, R., Lewis, J., Nauels, A., and Meinshausen, M.: Implications of non-linearities between cumulative CO2 emissions and CO2-induced warming for assessing the remaining carbon budget, Environmental Research Letters, 15, 074 017, 2020.

---

## Author Response (AR3)

Dear Prof. Kirk-Davidoff,

I wish to express my profound gratitude for recommending my manuscript titled 'Carbon Budget Concept and its Deviation Through the Pulse Response Lens' as a highlight in the 'Earth System Dynamics' journal. The article has improved significantly in the last two rounds of major revisions. Hence, I would like to thank you for your patience in guiding the revision process and being responsive to any clarifying questions that arose along the way. Moreover, thank you for your flexibility and support in granting an extension for the revision when it was needed.

Moreover, I am grateful to the reviewers for their highly constructive feedback and criticism, which helped shape the manuscript into its current form.

The final upload of the manuscript incorporates the technical corrections suggested by the second reviewer, who worked generously and assiduously to identify them. Indeed, I would like to thank her/him once again through this channel for their immense contribution to this paper.

The only correction that slightly stands out from their recommendation is that I chose to use the term a simple climate model (SCM) instead of a reduced-complexity model in most places (instead of a full model). The motivation was to avoid confusion, as Green's function could be arguably interpreted as a "reduced-complexity" model of an associated reduced-complexity model. In many places, I just stated "FaIR" instead.

Finally, the figures in the Supplement were already re-labeled in the previous iteration. Now, they stand as Figure S1 instead of Figure 1 (referring to the Notification to the authors in MS records).

Thank you again.
Best regards,
Vito Avakumović